# Stability beyond Bounded Differences: Sharp Generalization Bounds under Finite $L_p$ Moments

Qianqian Lei [1]    Soham Bonnerjee [1]    Yuefeng Han [2]    Wei Biao Wu [1]

## Abstract

While algorithmic stability is a central tool for understanding generalization of learning algorithms, existing high-probability guarantees typically rely on uniform boundedness or sub-Gaussian/sub-Weibull tail assumptions, which can be overly restrictive for modern settings with heavy-tailed or unbounded losses. We develop a stability-based framework that requires only a finite $L_p$ moment condition. Our first contribution is sharp concentration inequalities for functions of independent random variables under $L_p$ constraints, extending McDiarmid's bounded-differences techniques beyond the classical regime. Leveraging these results, we derive sharp high-probability generalization bounds across a range of learning paradigms, including empirical risk minimization, transductive regression, and meta-learning. These guarantees show that $L_p$ stability suffices for robust generalization even when boundedness fails, substantially weakening the standard assumptions in the stability literature.

## 1. Introduction

Algorithmic stability has emerged as a fundamental tool for the theoretical analysis of machine learning algorithms, providing a principled framework for establishing generalization bounds. Seminal work by Bousquet & Elisseeff (2002) established that uniform stability is sufficient to bound the discrepancy between empirical and true risk for symmetric learning algorithms. Unlike complexity-based measures such as VC-dimension (Vapnik & Chervonenkis, 1971; Blumer et al., 1989; Vapnik, 1991; 1999; Shalev-Shwartz et al., 2010; Shalev-Shwartz & Ben-David, 2014) and Rademacher complexity (Bartlett & Mendelson, 2003; Bartlett et al., 2005; van de Geer, 2006), stability analysis quantifies how perturbations in training data affect the output hypothesis, thereby providing insight into why machine learning algorithms trained on finite samples generalize to unseen data. Subsequent work has extended these ideas to broader settings, including stochastic optimization (Hardt et al., 2016; Kuzborskij & Lampert, 2018; Lei & Ying, 2020), federated learning (Sun et al., 2024; Liu et al., 2025), and reinforcement learning (Tamar et al., 2021; Smith & Whiteson, 2022). A significant portion of this literature derives high-probability control over generalization error via uniform stability (Bousquet & Elisseeff, 2002; Maurer, 2005; Warnke, 2016; Bousquet et al., 2020; Zhou et al., 2023) or its relaxations through distribution-dependent surrogates such as sub-Gaussian or sub-exponential conditions (Kontorovich, 2014; Maurer & Pontil, 2021; Li & Liu, 2023; Li et al., 2024; Fan & Lei, 2024; Han et al., 2023; 2026).

Though powerful, these approaches fail to capture the complexities of modern data science and deep learning. Recent empirical and theoretical studies (Catoni, 2012; Mendelson, 2014; Simsekli et al., 2019; Martin & Mahoney, 2021) highlight the inadequacy of sub-Gaussian assumptions for data with heavy tails. Instead, these studies reveal a prevalence of power law behavior–that is, polynomial tail in the large deviation regimes–reminiscent of the $t$-distribution. To accommodate such behavior, in this work, we introduce a further relaxation to $(L_p, \beta)$-*Lipschitz stability*, which requires only a finite $p$-th moment rather than uniform constant bounds or tail specifications such as sub-Gaussian or sub-Weibull conditions. Drawing on the classical Nagaev-type inequalities (Nagaev, 1979) as a powerful alternative when exponential concentration is unattainable, we derive novel sharp concentration bounds for generalization error and apply them to a range of settings, including empirical risk minimization, transductive learning, and meta-learning.

### 1.1. Main Contributions

Our contributions can be summarized as follows.

- In Section 2, we establish new *sharp* concentration inequalities for general functions of independent random

---

[1]Department of Statistics, The University of Chicago, Chicago, IL 60637, USA. [2]Department of Applied and Computational Mathematics and Statistics, University of Notre Dame, Notre Dame, IN 46556, USA. Correspondence to: Yuefeng Han <yuefeng.han@nd.edu>.

*Proceedings of the $43^{rd}$ International Conference on Machine Learning*, Seoul, South Korea. PMLR 306, 2026. Copyright 2026 by the author(s).

variables under $L_p$ moment. The resulting bounds exhibit a clear two-regime structure: a sub-Gaussian tail governing moderate deviations and a polynomial correction capturing rare large fluctuations. Informally:

**Theorem 1.1** (Theorem 2.2, informal). *Let $x_1, ..., x_n$ be independent random variables, and let $x_i'$ be an independent copy of $x_i$. Suppose there exist functions $f$ and $\{H_i\}_{i=1}^n$ such that, almost surely, $|f(x_1, \ldots, x_n) - f(x_1, \ldots, x_{i-1}, x_i', x_{i+1}, \ldots, x_n)| \leq H_i(x_i, x_i')$. If $H_i(x_i, x_i')$ has a finite p-th moment for $p \geq 2$, then for $y > 0$,*

$$\mathbb{P}(|f(x_1, \ldots, x_n) - \mathbb{E}[f(x_1, \ldots, x_n)]| > y)$$
$$\lesssim \frac{\sum_{i=1}^n \mathbb{E}[|H_i|^p]}{y^p} + \exp\left(-\frac{y^2}{\sum_{i=1}^n \mathbb{E}[|H_i|^2]}\right).$$

We also develop the result for the complementary heavy-tailed regime $p \in (1, 2)$ in Theorem 2.6. This two-regime behavior arises directly from working under weaker $L_p$ moment assumptions rather than bounded or sub-Weibull conditions, yielding non-asymptotic guarantees that remain valid without exponential moments.

- Using these probabilistic tools, we derive high-probability generalization bounds for three learning paradigms: empirical risk minimization (Section 3.1), transductive regression (Section 3.2), and meta-learning (Section 3.3). For each setting, we formulate an appropriate $(L_p, \beta)$-Lipschitz stability notion and obtain corresponding two-regime bounds. The transductive setting requires concentration inequalities for sampling without replacement, developed in Theorem 3.9. Compared to bounded uniform stability, the weaker $L_p$ framework relaxes tail assumptions but imposes stronger decay requirements on the relevant stability notion to achieve vanishing generalization bounds.

- Our numerical experiments confirm the sharpness of the theory and, in particular, highlight the necessity of the polynomial term in our generalization bounds. The results clearly exhibit the predicted transition between the sub-Gaussian and polynomial-tail regimes.

## 1.2. Related Work

**Algorithmic Stability and Generalization.** Algorithmic stability offers an algorithm-dependent route to generalization bounds, initiated by Bousquet & Elisseeff (2002) who combined uniform stability and bounded losses via McDiarmid's inequality (McDiarmid, 1989); later works sharpened rates (Feldman & Vondrak, 2018; 2019; Bousquet et al., 2020; Klochkov & Zhivotovskiy, 2021; Zhou et al., 2023) and connected stability to optimization (Hardt et al., 2016; Kuzborskij & Lampert, 2018; Lei & Ying, 2020). Since expectation bounds can mask non-negligible probabilities of large deviations for a single learned model, high-probability

guarantees are needed (Maurer, 2005; Bousquet et al., 2020; Yuan & Li, 2023a), but typically rely on bounded differences. For unbounded stability, existing approaches either tolerate rare large deviations under high-probability boundedness (Kutin, 2002; Kutin & Niyogi, 2002; Warnke, 2016), or replace worst-case bounds by distributional surrogates such as sub-Gaussian/sub-exponential assumptions (Kontorovich, 2014; Maurer & Pontil, 2021; Li & Liu, 2023; Li et al., 2024; Fan & Lei, 2024), often via Efron-Stein-type inequalities (Abou-Moustafa & Szepesvári, 2019), or provide moment bound (Yuan & Li, 2023b). Another route avoids exponential moments by robustifying empirical risk (or gradients) using robust mean estimators: Catoni-type M-estimators for robust ERM (Catoni, 2012; Brownlees et al., 2015), median-of-means extensions (Hsu & Sabato, 2016; Lecué & Lerasle, 2020), and truncation/clipping methods such as robust gradient descent (Holland & Ikeda, 2019), which can yield exponential-type deviation bounds under low-order moments but modify the learning rule. Our $L_p$ stability framework instead *relaxes tail assumptions directly*, accommodating heavy-tailed regimes where exponential moments may not exist, without changing the learning rule.

**Transductive Regression.** In transductive learning, the learner observes unlabeled test inputs in advance and predicts only on this fixed set (Vapnik, 2006). Cortes & Mohri (2006) provided systematic VC dimension bounds for transductive regression, and Cortes et al. (2008) derived algorithm-dependent generalization bounds via stability analysis. We extend this framework to the $L_p$ setting, requiring new concentration inequalities for sampling without replacement.

**Meta-Learning.** Meta-learning studies how training on past tasks improves performance on new tasks (Bengio et al., 1990; Thrun & Pratt, 1998; Maurer, 2005; Hoskins & Fredriksson, 2008). Theoretical work focuses on convergence (Zhou et al., 2019; Ji et al., 2020; Mishchenko et al., 2023) and generalization (Baxter, 2000; Ben-David & Schuller, 2003; Du et al., 2021); see Wang & Arora (2024) for a review. Stability-based analyses were initiated by Maurer (2005) and recently advanced by Chen et al. (2020); Fallah et al. (2021); Al-Shedivat et al. (2021); Guan et al. (2022) under various meta-stability notions. Wang & Arora (2024) proposed uniform meta-stability coupling task-level and within-task perturbations with high-probability bounds. Our $L_p$ framework generalizes these results under weaker moment assumptions.

## 2. Main Results

In this section, as a key technical contribution, we present sharp concentration inequalities that substantially broaden the scope of algorithmic stability. Classical high-probability stability bounds rely on uniform stability (Bousquet & Elis-

seeff, 2002), which requires uniform boundedness of the loss, or sub-Gaussian/sub-Weibull tail assumptions. We replace these restrictions with a single, weaker requirement: a finite $L_p$ moment for the one-sample perturbation. This relaxation is essential in modern applications, such as deep learning with heavy-tailed gradients or regression over unbounded domains, where the effect of replacing one training sample is not uniformly bounded but can be controlled in $L_p$ norm. Our framework thus provides a powerful theoretical tool for analyzing generalization in learning algorithms.

## 2.1. Preliminaries

We first introduce the mathematical framework used throughout the paper. Let $x_1, \ldots, x_n$ be independent random variables in a measurable space $\mathcal{X}$, and let $f : \mathcal{X}^n \to \mathbb{R}$ be a measurable function. We study the concentration of $f(x_1, \ldots, x_n)$ around its expectation under an $L_p$-type moment condition. Fix $i \in [n]$ and define $g = f(x_1, \ldots, x_i, \ldots, x_n)$ and $g_i = f(x_1, \ldots, x_i', \ldots, x_n)$, where $x_i'$ is an independent copy of $x_i$. Assume that for some nonnegative measurable functions $H_i : \mathcal{X}^2 \to \mathbb{R}_+$,

$$|g - g_i| \le H_i(x_i, x_i'). \tag{1}$$

Each $H_i$ is *deterministic*; randomness arises only through the pair $(x_i, x_i')$. We interpret $H_i$ as measuring the sensitivity to resampling the $i$-th coordinate: the one-point perturbation bound (1) is a Lipschitz-type condition with respect to this coordinate-wise distance, consistent with viewing stability assumptions as Lipschitz continuity in an appropriate metric structure (Li et al., 2024).

When $H_i$ is a metric (or is dominated by one), it induces the $\ell_1$ product metric on $\mathcal{X}^n$, $\rho^{(n)}(x, x') := \sum_{i=1}^n H_i(x_i, x_i')$, yielding a geometric interpretation: modifying a single coordinate incurs cost $H_i(x_i, x_i')$, while multi-coordinate perturbations accumulate additively, as for Lipschitz functions on product spaces (Kontorovich, 2014). In unbounded settings, $(\mathcal{X}, H_i)$ may have infinite diameter, so we instead quantify the *distribution-dependent scale* via an independent pair $(x_i, x_i')$:

$$\|g - g_i\|_p < \|H_i(x_i, x_i')\|_p < \infty.$$

This $L_p$-diameter is an analogue of Orlicz/sub-Weibull diameters, except we require only a finite $p$-th moment rather than a $\psi_\alpha$-norm bound (Kontorovich, 2014; Li et al., 2024); see also Remark 2.1. We refer to this as $L_p$-*Lipschitz stability*, which replaces bounded differences with an integrable, distribution-dependent surrogate.

*Remark* 2.1. The $\psi_\alpha$ Orlicz norm is defined as

$$\|X\|_{\psi_\alpha} = \inf\{\theta > 0 : E[e^{(|X|/\theta)^\alpha}] \le 2\}. \tag{2}$$

This class includes sub-exponential ($\alpha = 1$) and sub-Gaussian ($\alpha = 2$) variables, but still requires an exponential moment, implying that all $L_p$ moments exist and

scale as $O(p^{1/\alpha})$. In contrast, we assume only a finite $L_p$ moment for a fixed $p$, which is strictly weaker. For instance, heavy-tailed variables such as Pareto distributions with shape parameter $p$ can have finite $L_p$ moments yet fail every sub-Weibull condition due to the nonexistence of the moment-generating function. Therefore, our framework substantially weakens the tail requirements on the stability variable $H_i$, yielding high-probability guarantees in regimes where sub-Weibull-based stability is inapplicable.

## 2.2. Concentration Inequalities

With $L_p$-Lipschitz stability in place, we now state our main concentration bounds. We begin with the case $p \ge 2$, where the tail behavior is captured by a hybrid inequality combining a polynomial (moment) term with a sub-Gaussian term. This form is well suited to settings where the algorithm is typically stable but may exhibit occasional large deviations due to heavy-tailed data, contamination, or irregular loss landscapes.

**Theorem 2.2.** *Let $x_1, \ldots, x_n$ be independent random variables taking values in a measurable space $\mathcal{X}$, and let $f : \mathcal{X}^n \to \mathbb{R}$ be a measurable function such that for some nonnegative measurable functions $H_i : \mathcal{X}^2 \to \mathbb{R}_+$, (1) holds. If $\|H_i(x_i, x_i')\|_p < \infty$ for some $p \ge 2$ and all $i$, then for all $z > 0$,*

$$\mathbb{P}(|f(x_1, x_2, \ldots, x_n) - \mathbb{E}[f(x_1, x_2, \ldots, x_n)]| > z)$$
$$\le c_1 \frac{\sum_{i=1}^n \mathbb{E}|H_i(x_i, x_i')|^p}{z^p} + 2\exp\left\{-\frac{c_2 z^2}{\sum_{i=1}^n \mathbb{E}|H_i(x_i, x_i')|^2}\right\}, \tag{3}$$

*where $c_1 = 4(4\frac{p+2}{p})^p$ and $c_2 = (2(p+2)^2 e^p)^{-1}$ depend only on $p$.*

*Remark* 2.3. Theorem 2.2 admits an equivalent high-probability form. For any $\delta \in (0, 1)$, with probability at least $1 - \delta$,

$$\left|f(x_1, \ldots, x_n) - \mathbb{E}f(x_1, \ldots, x_n)\right|$$
$$\le c_2^{-1/2} \sqrt{\log\frac{4}{\delta}} \left(\sum_{i=1}^n \|H_i(x_i, x_i')\|_2^2\right)^{1/2}$$
$$+ (2c_1)^{1/p} \delta^{-1/p} \left(\sum_{i=1}^n \|H_i(x_i, x_i')\|_p^p\right)^{1/p}. \tag{4}$$

Using $\left(\sum_{i=1}^n \|H_i\|_p^p\right)^{1/p} \le n^{1/p} \max_i \|H_i\|_p$, the second term in (4) simplifies to a max-type bound:

$$\left|f(x_1, \ldots, x_n) - \mathbb{E}f(x_1, \ldots, x_n)\right|$$
$$\le c_2^{-1/2} \sqrt{\log\frac{4}{\delta}} \left(\sum_{i=1}^n \|H_i(x_i, x_i')\|_2^2\right)^{1/2}$$
$$+ (2c_1)^{1/p} n^{1/p} \delta^{-1/p} \max_{1 \le i \le n} \|H_i(x_i, x_i')\|_p.$$

The deviation thus decomposes into a sub-Gaussian term governed by $\sum_i \|H_i(x_i, x_i')\|_2^2$ and a heavy-tail correction controlled by the $p$-th moments.

*Remark* 2.4. A closely related result is Corollary 29 of Li & Liu (2024), whose statement shares the same Gaussian-plus-polynomial form as Theorem 2.2. However, its proof relies on a direct application of Markov's inequality, which is not sharp enough to establish the conclusion uniformly over all deviation levels $z > 0$ under a single fixed $p$-th moment condition. The issue can already be seen in the simplest example. Let $X_i \overset{\text{i.i.d.}}{\sim} N(0,1)$ and $f(X_1, \ldots, X_n) = \sum_{i=1}^{n} X_i$. A direct Markov argument based on the fixed $p$-th moment gives $\mathbb{P}(|\sum_{i=1}^{n} X_i| > t) \leq \frac{\mathbb{E}|\sum_{i=1}^{n} X_i|^p}{t^p} = \frac{\mathbb{E}|N(0,1)|^p \, n^{p/2}}{t^p}$, which is worse than the polynomial term in Theorem 2.2 by a factor $n^{p/2-1}$ for $p > 2$ and fails to recover the sub-Gaussian component entirely. In contrast, our proof employs a truncation and martingale argument to derive the Nagaev-type bound directly and uniformly over all $z > 0$.

*Remark* 2.5 (Sharpness). Throughout the paper, we use the term "sharp" in the tail-order sense, rather than in a minimax sense. For Theorem 2.2, sharpness refers to the precise large-deviation form under a single fixed finite-$L_p$ replace-one condition: the deviation probability consists of a sub-Gaussian moderate-deviation component and a polynomial large-deviation correction. The polynomial term is not an artifact of the proof; it reflects the unavoidable large-deviation behavior allowed by finite-moment increments.

This interpretation is consistent with the classical Fuk–Nagaev phenomenon for sums of independent heavy-tailed random variables, where the correct non-asymptotic tail form contains both a Gaussian term and a one-large-jump polynomial term. Exact moderate and large deviation results for linear processes further show that this two-regime structure can hold at the level of asymptotic equality, not merely as an upper-bound phenomenon; see, for instance, Peligrad et al. (2014). Our theorem extends this tail-order structure from explicit sums to general functions satisfying a replace-one $L_p$ envelope condition. The bounds in Section 3 are sharp in the same sense: once the corresponding stability assumption reduces the generalization gap to Theorem 2.2, the resulting high-probability bounds inherit this Gaussian-plus-polynomial tail-order structure.

We also provide an accompanying result when only lower moments exist.

**Theorem 2.6.** *Let $x_1, \ldots, x_n$ be independent random variables taking values in a measurable space $\mathcal{X}$, and let $f : \mathcal{X}^n \to \mathbb{R}$ be a measurable function. For each $i \in [n]$, define $g = f(x_1, \ldots, x_{i-1}, x_i, x_{i+1}, \ldots, x_n)$ and $g_i = f(x_1, \ldots, x_{i-1}, x_i', x_{i+1}, \ldots, x_n)$, where $x_i'$ is an independent copy of $x_i$. Assume there exist nonnegative measurable functions $H_i : \mathcal{X}^2 \to \mathbb{R}_+$, such that*

$$|g - g_i| \leq H_i(x_i, x_i'), \quad i = 1, \ldots, n,$$

*with $\|H_i(x_i, x_i')\|_p < \infty$ for some $1 < p < 2$ and all $i$.*

*Then, for all $Q \geq 1$ and $z > 0$,*

$$\mathbb{P}\left(|f(x_1, \ldots, x_n) - \mathbb{E}f(x_1, \ldots, x_n)| > z\right)$$

$$\leq \sum_{i=1}^{n} \mathbb{P}(|H_i(x_i, x_i')| > \frac{z}{4Q})$$

$$+ 2 \left( \frac{e 4^p Q^{p-1} \sum_{i=1}^{n} \mathbb{E}|H_i(x_i, x_i')|^p}{z^p} \right)^Q$$

*Remark* 2.7. Theorem 2.6 is essentially sharp for polynomially-tailed variables. Consider $n = 1$ with $H_1(x_1, x_1') \sim |t_2|$ (Student's $t$ with $\nu = 2$ degrees of freedom). Then $\|H_1(x_1, x_1')\|_p < \infty$ for every $1 < p < 2$.

The $t_2$ distribution has the closed-form tail

$$\mathbb{P}(|H_1(x_1, x_1')| > \frac{z}{4Q}) = 1 - \frac{z/4Q}{\sqrt{z^2/16Q^2 + 2}} \sim \frac{1}{z^2} \ (z \to \infty),$$

so the first term in the bound is of order $z^{-2}$. For any fixed $Q > 2/p$, the second term satisfies $O(z^{-pQ}) = o(z^{-2})$. Hence the bound yields an $O(z^{-2})$ tail rate, matching the true exponent of the $t_2$ distribution.

In particular, under only an $L_p$ increment condition with $p < 2$, one cannot generally expect sub-Gaussian or sub-exponential decay. Compared with the naive Markov bound based solely on $\|H_1\|_p$, which gives order $z^{-p}$, Theorem 2.6 recovers the correct $z^{-2}$ rate in this canonical heavy-tailed example.

*Remark* 2.8. Table 1 compares our results with representative concentration inequalities under assumptions ranging from bounded, sub-Gaussian, and sub-exponential to sub-Weibull coordinate diameters.

Theorem 2.2 yields a hybrid bound combining a sub-Gaussian component and a polynomial term. For moderate deviations (small $z$), the sub-Gaussian term in (3) dominates, matching McDiarmid-type tail behavior and agreeing with Kontorovich (2014); Maurer & Pontil (2021); Li et al. (2024) up to constants. For large deviations (large $z$), the polynomial term dominates, reflecting that our assumptions impose only finite $L_p$ moments; the tail therefore decays polynomially rather than sub-Weibull. Our simulations illustrate this transition between the sub-Gaussian and heavy-tail regimes in two application settings.

## 3. Applications

### 3.1. Empirical Risk Minimization (ERM)

In this subsection, we illustrate how our concentration results yield high-probability generalization bounds in the standard i.i.d. setting. Let $\mathcal{X}$ and $\mathcal{Y}$ be the input and output spaces, and let $\mathcal{Z} = \mathcal{X} \times \mathcal{Y}$. Consider a training sample $S = \{z_1 = (x_1, y_1), \ldots, z_m = (x_m, y_m)\} \sim \mathcal{D}^m$, drawn i.i.d. from an unknown distribution $\mathcal{D}$. A learning algorithm is a map $A : \mathcal{Z}^m \to \mathcal{F}$, producing a hypothesis

*Table 1.* Comparison with prior concentration results. For simplicity, we write $H_i := H_i(x_i, x_i')$ and focus on the relevant Orlicz norm.

| Result | Assumption | Deviation bound |
|---|---|---|
| Kontorovich (2014) | $f$ is 1-Lipschitz; $\|H_i\|_{\psi_2} < \infty$ for all $i$ | $\mathbb{P}(|f - \mathbb{E}f| > z) \le 2\exp\left(-\dfrac{z^2}{2\sum_{i=1}^n \|H_i\|_{\psi_2}^2}\right)$ |
| Maurer & Pontil (2021) | $f$ is 1-Lipschitz; $\|H_i\|_{\psi_1} < \infty$ for all $i$ | $\mathbb{P}(f - \mathbb{E}f > z) \le \exp\left(-\dfrac{z^2}{4e^2\sum_{i=1}^n \|H_i\|_{\psi_1}^2 + 2e(\max_i \|H_i\|_{\psi_1})z}\right)$ |
| Li et al. (2024) | $|f - f_i| \le H_i$, $\|H_i\|_{\psi_\alpha} < \infty$, $0 < \alpha \le 1$ | $\mathbb{P}(|f - \mathbb{E}f| > z) \le \exp\left(-c_\alpha \dfrac{z^2}{\sum_{i=1}^n \|H_i\|_{\psi_\alpha}^2}\right) + \exp\left(-\dfrac{z^\alpha}{\max_i \|H_i\|_{\psi_\alpha}^\alpha}\right)$ |
| Li et al. (2024) | $|f - f_i| \le H_i$, $\|H_i\|_{\psi_\alpha} < \infty, \alpha > 1$ and $\alpha^{-1} + (\alpha^*)^{-1} = 1$ | $\mathbb{P}(|f - \mathbb{E}f| > z) \le \exp\left(-c_\alpha \dfrac{z^2}{\sum_{i=1}^n \|H_i\|_{\psi_\alpha}^2}\right) + \exp\left(-\dfrac{z^\alpha}{\left(\sum_{i=1}^n \|H_i\|_{\psi_\alpha}^{\alpha^*}\right)^{\alpha/\alpha^*}}\right)$ |
| **Ours (Thm. 2.2)** | $|f - f_i| \le H_i$, $\|H_i\|_p < \infty$ for some $p \ge 2$ | $\mathbb{P}(|f - \mathbb{E}f| > z) \le c_{1,p} \dfrac{\sum_{i=1}^n \mathbb{E}|H_i|^p}{z^p} + 2\exp\left(-c_{2,p}\dfrac{z^2}{\sum_{i=1}^n \mathbb{E}|H_i|^2}\right)$ |
| **Ours (Thm. 2.6)** | $|f - f_i| \le H_i$, $\|H_i\|_p < \infty$ for some $1 < p < 2$ | $\mathbb{P}(|f - \mathbb{E}f| > z) \le \sum_{i=1}^n \mathbb{P}(|H_i| > \frac{z}{4Q}) + 2\left(\dfrac{e^{4p}Q^{p-1}\sum_{i=1}^n \mathbb{E}|H_i|^p}{z^p}\right)^Q$ |

$A_S \in \mathcal{F} \subseteq \mathcal{Y}$. For simplicity, we assume $A$ is deterministic and permutation-invariant in $S$. We also adopt standard measurability assumptions.

- $S^{\setminus i}$, the leave-one-out sample of size $m - 1$ obtained by removing the $i$-th observation:

$$S^{\setminus i} = S \setminus z_i = \{z_1, \ldots, z_{i-1}, z_{i+1}, \ldots, z_m\}.$$

- $S^i$, the replacement sample of size $m$ where $z_i$ is replaced by an independent draw $z_i' \sim \mathcal{D}$:

$$S^i = \{z_1, \ldots, z_{i-1}, z_i', z_{i+1}, \ldots, z_m\}.$$

All expectations and probabilities are taken with respect to the data distribution $\mathcal{D}$. We use subscripts to specify the variables of integration: $\mathbb{E}_S[\cdot]$ denotes the expectation over the training sample $S \sim \mathcal{D}^m$, while $\mathbb{E}_z[\cdot]$ denotes the expectation over a single test instance $z \sim \mathcal{D}$.

To measure the discrepancy between a prediction $f(x)$ and the ground truth $y$, we use a nonnegative cost function $c : \mathcal{Y} \times \mathcal{Y} \to \mathbb{R}_+$. For any hypothesis $f$ and sample $z = (x, y)$, define the loss $\ell(f, z) = c(f(x), y)$. Given a training sample $S$, the population risk of the learned hypothesis $A_S$ is $R(A, S) = \mathbb{E}_z[\ell(A_S, z)]$, where $z \sim \mathcal{D}$ is an independent test point. Since $\mathcal{D}$ is unknown, $R(A, S)$ cannot be computed directly. We therefore compare it to empirical surrogates: the empirical risk and leave-one-out risk, defined respectively as

$$R_{emp}(A, S) = \frac{1}{m} \sum_{i=1}^m \ell(A_S, z_i),$$

$$R_{loo}(A, S) = \frac{1}{m} \sum_{i=1}^m \ell(A_{S^{\setminus i}}, z_i).$$

When the algorithm $A$ and sample $S$ are clear from context, we write simply $R$, $R_{emp}$, and $R_{loo}$.

The classical route to high-probability generalization bounds typically relies on deterministic *uniform stability*: the point-wise change in loss under a one-sample perturbation is assumed to be bounded, often together with an almost surely bounded loss. These conditions make McDiarmid-type concentration inequalities straightforward, but they are too restrictive for many modern problems, such as regression with unbounded responses or heavy-tailed noise. Leveraging Theorem 2.2, we instead adopt an $(L_p, \beta)$-Lipschitz stability framework.

**Assumption 3.1** (($L_p, \beta)$-Lipschitz stability). An algorithm $A$ trained on set $S = \{z_1, \ldots, z_m\}$ is $\beta$-Lipschitz stable if the function $f(z_1, \ldots, z_m, z) = \ell(A_S, z)$ with respect to the loss $\ell$ is $\beta$-Lipschitz, and with respect to a measurable function $H : \mathcal{Z} \times \mathcal{Z} \mapsto \mathbb{R}_+$ such that for all $i \in \{1, \ldots, m\}$,

$$|\ell(A_S, z) - \ell(A_{S^i}, z)| \le \beta H(z_i, z_i'),$$

where $\|H(z, z')\|_p < \infty$ for some $p \ge 2$, and the norm $\|\cdot\|_p$ is computed with respect to $z \sim \mathcal{D}$.

Assumption 3.1 significantly generalizes the Lipschitz stability notion of Kontorovich (2014); Li et al. (2024), and consequently also generalizes the uniform stability of Bousquet & Elisseeff (2002), which in turn implies hypothesis stability. It can also be construed as generalizing the notions of argument stability (Liu et al., 2017; Wang et al., 2022), random uniform stability (Shen et al., 2020).

*Remark* 3.2. Assumption 3.1 is compatible with, and strictly extends, several standard stability regimes. If the replace-one stability increment is uniformly bounded, one may take $H \equiv 1$, recovering the classical deterministic uniform-stability setting. This includes standard uniformly stable regularized ERM procedures, such as Hilbert-space regularization and support vector machines under bounded-loss assumptions. Similarly, if the stability envelope $H$ is sub-Gaussian, sub-exponential, or sub-Weibull, then $H$ has finite moments of all orders, and Assumption 3.1 holds for any fixed finite $p$. Our framework therefore does not replace these light-tailed assumptions where they apply; rather, it

covers the complementary regime in which exponential-tail assumptions are unavailable but a finite $p$-th moment exists.

Random, sample-dependent stability/Lipschitz envelopes arise naturally in modern learning settings. Kontorovich (2014) studies stability in unbounded metric spaces and gives generalization bounds for unbounded losses, including regularized metric-regression algorithms. In private optimization, Das et al. (2023) replace uniform Lipschitzness in DP-SGD by sample-dependent per-example Lipschitz constants with bounded moments, with applications to private softmax-layer training. The SGD stability literature (Hardt et al., 2016) similarly shows that stochastic gradient methods can be stable under smoothness and Lipschitz-type conditions. While these works do not verify Assumption 3.1 for every modern optimizer, they show that finite-moment, sample-dependent perturbation scales are natural; once such an $L_p$ replace-one envelope is established, Theorem 3.4 gives the corresponding high-probability generalization bound.

**Assumption 3.3.** There exists a measurable function $G : \mathcal{Z} \times \mathcal{Z} \mapsto \mathbb{R}_+$, such that for all $S = \{z_1, \ldots, z_m\} \in \mathcal{D}^m$, $z, z' \sim \mathcal{D}$,

$$|\ell(A_S, z) - \ell(A_S, z')| \leq \beta' G(z, z'), \ \beta' > 0,$$

where $\|G(z, z')\|_p < \infty$ for some $p \geq 2$, and the norm $\|\cdot\|_p$ is computed with respect to $z, z' \sim \mathcal{D}^2$.

Combining these assumptions with Theorem 2.2 yields the following high-probability generalization bounds.

**Theorem 3.4.** *Suppose learning algorithm $A$ satisfies Assumption 3.1 and the loss function $\ell(A_S, z)$ satisfies Assumption 3.3. Then, for any set $S$ with $|S| = m \geq 1$ and any $y > 0$, the following two bounds hold:*

$$\mathbb{P}(|R - R_{emp}| > y + \beta\mathbb{E}H(z, z'))$$
$$\leq c_1 \frac{m\beta^p\mathbb{E}[|H(z, z')|^p] + (\beta')^p\mathbb{E}[|G(z, z')|^p]/m^{p-1}}{y^p}$$
$$+ 2\exp\left(-\frac{c_2 y^2}{\beta^2 m\mathbb{E}|H(z, z')|^2 + \frac{(\beta')^2}{m}\mathbb{E}[|G(z, z')|^2]}\right),$$
$$\mathbb{P}(|R - R_{loo} - (R_m - R_{m-1})| > y)$$
$$\leq c_3 \frac{m\beta^p\mathbb{E}[|H(z, z')|^p] + (\beta')^p\mathbb{E}[|G(z, z')|^p]/m^{p-1}}{y^p}$$
$$+ 2\exp\left(-\frac{c_4 y^2}{m\beta^2\mathbb{E}[|H(z, z')|^2] + \frac{(\beta')^2}{m}\mathbb{E}[|G(z, z')|^2]}\right),$$

*where $c_1, c_2, c_3, c_4$ depend only on $p$, and $R_m := \mathbb{E}_{S \sim \mathcal{D}^m, z \sim \mathcal{D}}[\ell(A_S, z)] = \mathbb{E}_{S \sim \mathcal{D}^m}[R(A, S)]$.*

*Remark* 3.5. We briefly recall classical generalization bounds for uniformly stable algorithms. A sharp high-probability result is Corollary 8 of Bousquet et al. (2020),

which assumes deterministic uniform stability with constant $\beta$ (i.e., Assumption 3.1 with $H \equiv 1$) and almost sure bounded loss $0 \leq \ell \leq L$. It states that, for any $\delta \in (0, 1)$, with probability at least $1 - \delta$,

$$|R - R_{emp}| \lesssim \beta \log m \log(1/\delta) + L\sqrt{\frac{\log(1/\delta)}{m}}.$$

This bound is informative only when $\beta = o(1)$; moreover, it is tight (up to logarithmic factors) in the regime $\beta \lesssim m^{-1/2}$, which yields the canonical $m^{-1/2}$ rate. In contrast, Theorem 3.4 allows the stability increment to be random and data-dependent (through $H(z_i, z_i')$) and does not require bounded loss; it suffices that $H$ and $G$ have finite $p$-th moments for some $p \geq 2$. In particular, with probability at least $1 - \delta$,

$$|R - R_{emp}| \lesssim (\beta\|H\|_p m^{1/p} + \beta'\|G\|_p m^{-(1-1/p)})\delta^{-1/p}$$
$$+ \left(\beta\|H\|_2\sqrt{m} + \beta'\frac{\|G\|_2}{\sqrt{m}}\right)\log\frac{1}{\delta} + \beta\|H\|_1.$$

For the bound to vanish, one needs $\|H\|_p = O(1)$, $\|G\|_p = O(1)$ and (up to logarithmic factors) $\beta \ll m^{-1/2}$, $\beta' \ll \sqrt{m}$; smaller $\delta$ further strengthens the required decay in $\beta, \beta'$ through the factor $\delta^{-1/p}$. This requirement is standard rather than a weakness of our proof: a closely related bound in Li et al. (2024) also contains a term of order $\gamma\Delta_\alpha\sqrt{m\log(1/\delta)}$, and the paper explicitly notes that the stability coefficient is typically of order $m^{-1/2}$. In this sense, moving from bounded uniform stability to weaker $(L_p, \beta)$-Lipschitz stability trades assumptions for stronger Lipschitz-decay requirements.

This condition is verifiable in standard examples. For instance, in ridge regression with squared loss, on the usual well-conditioned covariance event, the replace-one perturbation yields $\beta = O(\log m/m)$, and one may take $H(z, z') = \|xy - x'y'\|_2$. Thus the term $\beta\|H\|_2\sqrt{m}$ is controllable without imposing almost-sure boundedness on the covariates, responses, or losses. Appendix D.3 further provides an empirical illustration for a two-layer neural network trained with Adam, showing the same ratio-based tail behavior in a less structured learning algorithm.

### 3.2. Transductive Regression Algorithms

We consider the transductive learning setting (Cortes et al., 2008), where the learner is given a fixed finite population $\mathcal{X}$ of size $N = m + u$. A training set $S$ of $m$ labeled samples is drawn uniformly at random *without replacement* from $\mathcal{X}$; the remaining $u$ points form the unlabeled test set $T := \mathcal{X} \backslash S$. We denote this random partition by $\mathcal{X} \vdash (S, T)$.

The goal is to predict the labels of the test points in $T$ using only the labeled data in $S$. Unlike inductive learning, which aims to learn a function that generalizes to arbitrary future

inputs, transductive learning only targets a fixed, known set of test inputs. Access to the unlabeled set $T$ during training allows the algorithm to exploit the geometry or manifold structure of the test set to regularize learning and improve prediction.

Let $\ell(h, z) \geq 0$ be a nonnegative loss measuring the error of a hypothesis $h$ on sample $z = (x, y)$; for regression, a canonical choice is the squared loss $\ell(h, z) = (h(x) - y)^2$. Define the training and test (transductive) risks by

$$\hat{R}(h) = \frac{1}{m} \sum_{z \in S} \ell(h, z), \quad R(h) = \frac{1}{u} \sum_{z \in T} \ell(h, z). \quad (5)$$

Our goal is to control the generalization gap $R(h) - \hat{R}(h)$ via stability properties of the algorithm. Classical analyses often assume uniform $\beta$-stability, which imposes bounded differences uniformly over all partitions. To accommodate heavy-tailed losses or unbounded domains, we instead adopt an $(L_p, \beta)$-Lipschitz stability notion.

**Assumption 3.6** (Transductive $(L_p, \beta)$-Lipschitz stability). Let $A$ be a transductive learning algorithm. For a partition $\mathcal{X} \vdash (S, T)$, let $h$ be the hypothesis returned by $A$, and let $h'$ be the hypothesis returned for a modified partition $\mathcal{X} \vdash (S', T')$. We say $A$ is uniformly $L_p$-Lipschitz stable with respect to the cost function $\ell$ if there exist nonnegative measurable functions $H$ such that, whenever $(S', T')$ is obtained from $(S, T)$ by swapping exactly one point $x_i \in S$ with one point $x_{m+j} \in T$, then for all $x \in \mathcal{X}$,

$$|\ell(h, x) - \ell(h', x)| \leq \beta H(x_i, x_{m+j}),$$

and $\|H(x_i, x_{m+j})\|_p < \infty$ for some $p \geq 2$.

*Remark* 3.7. As with Assumption 3.1, we do not require a deterministic control. To see this, view $\ell(h, x) = f(x_1, \ldots, x_m, x)$ and $\ell(h', x) = f(x_1, \ldots, x_{i-1}, x_{m+j}, x_{i+1}, \ldots, x_m, x)$; the assumption above is then equivalent to the function $f : \mathcal{X}^{m+1} \to \mathbb{R}$ being $(L_p, \beta)$-Lipschitz Stable.

**Assumption 3.8** ($L_p$-bounded hypothesis class). A hypothesis class $\mathcal{H}$ is $L_p$-bounded with respect to $\ell$ if there exists a measurable function $G$ such that for all $h \in \mathcal{H}$ and all $x, x' \in \mathcal{X}$,

$$|\ell(h, x) - \ell(h, x')| \leq \beta' G(x, x'), \quad \beta' > 0,$$

where $\|G(x, x')\|_p < \infty$ for some $p \geq 2$, and the norm $\|\cdot\|_p$ is computed with respect to $x, x' \sim \mathcal{D}$.

A key technical challenge in transduction is that $S = \{x_1, \ldots, x_m\}$ is sampled without replacement from the finite population $\mathcal{X}$, which induces dependencies among the training points and precludes applying concentration inequalities for independent variables directly. Prior work addresses this using McDiarmid-type inequalities tailored

to sampling without replacement, typically under strict bounded-differences assumptions. In our $(L_p, \beta)$-Lipschitz stability regime, we develop corresponding extensions of Theorem 2.2 that accommodate the transductive sampling dependence while requiring only finite $L_p$ moments.

**Theorem 3.9.** *Let $X$ be a finite set with $|X| = N = m + u$. Let $x_1^m = (x_1, \ldots, x_m)$ be sampled uniformly without replacement from $X$ and let $\phi : X^m \to \mathbb{R}$ be a symmetric measurable function. Assume that for each $i \in [m]$ there exists a measurable function $H : X \times X \to \mathbb{R}$ such that for all $(x_1, \ldots, x_m) \in X^m$ and all $x_i' \in X$, where $x_i'$ is an independent copy of $x_i$, defining $\varphi = \phi(x_1, \ldots, x_{i-1}, x_i, x_{i+1}, \ldots, x_m)$ and $\varphi_i = \phi(x_1, \ldots, x_{i-1}, x_i', x_{i+1}, \ldots, x_m)$, we have*

$$|\varphi - \varphi_i| \leq H(x_i, x_i').$$

*Suppose that for some $p \geq 2$, $\|H(x, x')\|_p < \infty$ for all $i \in [m]$. Then for all $y > 0$,*

$$\mathbb{P}(|\varphi - \mathbb{E}\varphi| > y) \leq c_1 \frac{V_p}{y^p} + 2 \exp\left(-c_2 \frac{y^2}{V_2}\right), \quad (6)$$

*where*

$$V_p = \frac{u^p}{p-1}\left(\frac{1}{(u - \frac{1}{2})^{p-1}} - \frac{1}{(m + u - \frac{1}{2})^{p-1}}\right)\mathbb{E}|H(x, x')|^p,$$

$$V_2 = \frac{mu}{(m + u - 1/2)(1 - 1/(2\max\{m, u\}))}\mathbb{E}|H(x, x')|^2,$$

*and $c_1 = 4(4\frac{p+2}{p})^p$ and $c_2 = 1/(2(p+2)^2 e^p)$ are some constants depending only on $p$.*

We now apply Theorem 3.9 to study stability-based generalization for transductive regression. Our target is the generalization gap between the transductive risk and the training risk, which is defined as $\phi(S) := R(S) - \hat{R}(S)$. By controlling $|\mathbb{E}\phi(S)|$ and the one-swap increment $|\phi(S) - \phi(S')|$, where $S$ and $S'$ differ in exactly one point, we can invoke Theorem 3.9 to conclude the following result.

**Theorem 3.10.** *Let $\mathcal{H}$ be an $L_p$-bounded hypothesis class and let $A$ be a symmetric transductive $(L_p, \beta)$-Lipschitz stability algorithm satisfying Assumption 3.6. Let $h$ be the hypothesis returned by $A$ for a random partition $X \vdash (S, T)$. Then for any $y > 0$, we have*

$$\mathbb{P}\big(R(h) - \hat{R}(h) \geq y + \beta\mathbb{E}H(x_i, x_i')\big)$$
$$\leq \frac{c_1 V_p'}{y^p} + \exp\left(-\frac{c_2 y^2}{V_2'}\right),$$

*where*

$$V_p' = \frac{u^p}{p-1}\left(\frac{1}{(u - 1/2)^{p-1}} - \frac{1}{(m + u - 1/2)^{p-1}}\right)$$
$$\times \mathbb{E}|2\beta H(x, x') + |1/u - 1/m|\beta' G(x, x')|^p,$$

$$V_2' = \frac{mu}{(m + u - 1/2)(1 - 1/(2\max\{m, u\}))}$$
$$\times \mathbb{E}|2\beta H(x, x') + |1/u - 1/m|\beta' G(x, x')|^2,$$

*and the constants $c_1 = 4(4\frac{p+2}{p})^p$ and $c_2 = 1/(2(p+2)^2 e^p)$ depend only on $p$.*

*Remark* 3.11. In Appendix § C.2, we extend Theorems 3.9 and 3.10 to allow coordinate-dependent control through functions $H_i(x_i, x_i')$ as in Assumption 3.6. Theorem 3.10 shows that, with probability at least $1 - \delta$,

$$R(h) - \hat{R}(h)$$
$$\lesssim \sqrt{um}\Big(\beta\|H\|_2 + \frac{|u-m|}{um}\beta'\|G\|_2\Big)\sqrt{\log\frac{1}{\delta}}$$
$$+ (um)^{\frac{1}{p}}\Big(\beta\|H\|_p + \frac{|u-m|}{um}\beta'\|G\|_p\Big)\delta^{-\frac{1}{p}} + \beta\|H\|_1.$$

For comparison, under uniform stability with constant $\beta$ and almost surely bounded loss $0 \le \ell \le L$, Cortes et al. (2008) show that with probability at least $1 - \delta$,

$$R(h) - \hat{R}(h) \lesssim \beta + \Big(\beta + \frac{L^2(m+u)}{mu}\Big)\sqrt{m}\sqrt{\log\frac{1}{\delta}}.$$

To achieve vanishing bounds, their result, with significantly stronger assumptions, requires $\beta \ll m^{-1/2}$, whereas ours requires faster decay $\beta \ll (um)^{-1/2}$ with $\|H\|_p = O(1)$.

### 3.3. Meta-Learning

Consider a (possibly randomized) meta-learning algorithm $A$ acting on a meta-sample $\mathbb{S} = \{\mathcal{S}_1, \ldots, \mathcal{S}_m\}$, where each task dataset $\mathcal{S}_j = \{z_j^1, \ldots, z_j^n\}$ is drawn independently. Specifically, task distributions $\mathcal{D}_1, \ldots, \mathcal{D}_m$ are sampled i.i.d. from an unknown meta-distribution $\mu$ over a measurable space $\mathcal{Z}$, and for each $j$, the samples $z_j^1, \ldots, z_j^n$ are drawn i.i.d. from $\mathcal{D}_j$. Given $\mathbb{S}$, the meta-learner outputs a task-level learning algorithm $A(\mathbb{S})$, which, when trained on a new task dataset $\mathcal{S} \sim \mathcal{D}^n$ with $\mathcal{D} \sim \mu$, produces a model $A(\mathbb{S})(\mathcal{S}) \in \mathcal{P}$, where $\mathcal{P}$ denotes the model space. We evaluate a model $P \in \mathcal{P}$ on a test point $z \in \mathcal{Z}$ via a loss function $\ell : \mathcal{P} \times \mathcal{Z} \to \mathbb{R}_+$.

In the context of meta-learning, the empirical meta-risk of a meta-learning algorithm $A$, evaluated on the meta-sample $\mathbb{S}$, is given by

$$R(A(\mathbb{S}), \mathbb{S}) = \frac{1}{mn}\sum_{j=1}^{m}\sum_{i=1}^{n}\ell(A(\mathbb{S})(\mathcal{S}_j), z_j^i), \quad (7)$$

and the population meta-risk is

$$R(A(\mathbb{S}), \mu) = \mathbb{E}_{\mathcal{D}\sim\mu}\mathbb{E}_{(\mathcal{S},z)\sim\mathcal{D}^{n+1}}\ell(A(\mathbb{S})(\mathcal{S}), z). \quad (8)$$

In order to relate the empirical meta-risk (7) and the population meta-risk (8), we impose meta-stability assumptions. We first introduce neighboring datasets. For a task dataset $\mathcal{S} = \{z^1, \ldots, z^n\} \sim \mathcal{D}^n$ and index $i \in [n]$, let $\mathcal{S}^{(i)} = \{z^1, \ldots, z^{i-1}, z^{i'}, z^{i+1}, \ldots, z^n\}$, where $z^{i'} \sim \mathcal{D}$

is an independent copy of $z^i$. For a meta-sample $\mathbb{S}$ and indices $j \in [m], i \in [n]$, define the neighboring meta-sample $\mathbb{S}^{(j,i)} = \{\mathcal{S}_1, \ldots, \mathcal{S}_{j-1}, \mathcal{S}_j^{(i)}, \mathcal{S}_{j+1}, \ldots, \mathcal{S}_m\}$, so that $\mathbb{S}$ and $\mathbb{S}^{(j,i)}$ differ only by replacing $z_j^i$ with an i.i.d. copy $z_j^{i'} \sim \mathcal{D}_j$. We write $\mathcal{S}^{\backslash i} = \mathcal{S} \setminus \{z^i\}$ and $\mathbb{S}^{\backslash j} = \mathbb{S} \setminus \mathcal{S}_j$.

**Assumption 3.12.** The meta-learning algorithm $A$ satisfies the following three stability conditions.

(i) *Meta-stability across training tasks.* There exists a measurable function $H : \mathcal{Z} \times \mathcal{Z} \to \mathbb{R}_+$ with $\mathbb{E}_{\mathcal{D}\sim\mu}\mathbb{E}_{z,z' \overset{i.i.d.}{\sim} \mathcal{D}}|H(z,z')|^p < \infty$ for some $p \ge 2$ such that for all $j \in [m]$ and $i \in [n]$,

$$|\ell(A(\mathbb{S})(\mathcal{S}), z) - \ell(A(\mathbb{S}^{(j,i)})(\mathcal{S}), z)| \le \beta H(z_j^i, z_j^{i'}), \quad (9)$$

(ii) *Within-task stability.* There exists a measurable function $G : \mathcal{Z} \times \mathcal{Z} \to \mathbb{R}_+$ with $\mathbb{E}_{\mathcal{D}\sim\mu}\mathbb{E}_{z,z' \overset{i.i.d.}{\sim} \mathcal{D}}|G(z,z')|^{p'} < \infty$ for some $p' > 2$ such that for all $i \in [n]$,

$$|\ell(A(\mathbb{S})(\mathcal{S}), z) - \ell(A(\mathbb{S})(\mathcal{S}^{(i)}), z)| \le \beta' G(z^i, z^{i'}), \quad (10)$$

(iii) *Test-sample stability.* There exists a measurable function $\mathcal{M} : \mathcal{Z} \times \mathcal{Z} \to \mathbb{R}_+$ with $\mathbb{E}_{\mathcal{D}\sim\mu}\mathbb{E}_{z,z' \overset{i.i.d.}{\sim} \mathcal{D}}|\mathcal{M}(z,z')|^{p''} < \infty$ for some $p'' > 2$ such that for all $i \in [n]$,

$$|\ell(A(\mathbb{S})(\mathcal{S}), z) - \ell(A(\mathbb{S})(\mathcal{S}), z')| \le \beta''\mathcal{M}(z, z'), \quad (11)$$

Without loss of generality, we take $p' = p'' = p$; otherwise, one can work with $p \wedge p' \wedge p''$.

*Remark* 3.13. We contrast Assumption 3.12 with the closely related stability assumptions of Maurer (2005). First, Assumption 3.12 replaces almost-sure bounded stability gaps by a finite $L_p$-moment requirement. More importantly, part (i) is strictly weaker: Maurer (2005) perturbs the meta-sample by replacing an entire task with an i.i.d. copy, whereas we replace only a single within-task sample among the $m$ tasks in $\mathbb{S}$. This smaller, more realistic perturbation yields a more verifiable stability condition, yet remains sufficient for sharp high-probability bounds.

**Theorem 3.14.** *Let $\mu$ denote the task distribution. Given a meta-sample $\mathbb{S}$ and a meta-algorithm $A$, recall the empirical and population meta-risks $R(A(\mathbb{S}), \mathbb{S})$ and $R(A(\mathbb{S}), \mu)$ from (7) and (8) respectively. Suppose $A$ satisfies Assumption 3.12. Then, for all $y > 0$ it follows that*

$$\mathbb{P}\Big(R(A(\mathbb{S}), \mathbb{S}) - R(A(\mathbb{S}), \mu) \ge y + E[\beta H + \beta' G]\Big)$$
$$\le c_1 \frac{mnE|\beta H|^p + nm^{-(p-1)}E|\beta' G|^p + (mn)^{-(p-1)}E|\beta''\mathcal{M}|^p}{y^p}$$
$$+ 2\exp\Big(-\frac{c_2 y^2}{mnE|\beta H|^2 + nm^{-1}E|\beta' G|^2 + (mn)^{-1}E|\beta''\mathcal{M}|^2}\Big)$$

*where $\mathbb{E}$ is taken with respect to $z, z' \overset{i.i.d.}{\sim} \mathcal{D}$, $\mathcal{D} \sim \mu$, and $c_1, c_2$ are constants solely depending on $p$.*

*Remark* 3.15. Theorem 3.14 admits an equivalent high-probability form: for any $\delta \in (0,1)$, with probability at least $1 - \delta$,

$$R(A(\mathbb{S}), \mathbb{S}) - R(A(\mathbb{S}), \mu) \lesssim \mathbb{E}[\beta H + \beta' G]$$
$$+ \delta^{-\frac{1}{p}}\big((mn)^{\frac{1}{p}}\beta\|H\|_p + n^{\frac{1}{p}}m^{\frac{1}{p}-1}\beta'\|G\|_p\big)$$
$$+ \delta^{-\frac{1}{p}}\big((mn)^{\frac{1}{p}-1}\beta''\|\mathcal{M}\|_p\big) + \sqrt{\log(1/\delta)}\big(\sqrt{mn}\beta\|H\|_2\big)$$
$$+ \sqrt{\log(1/\delta)}\big(n^{\frac{1}{2}}m^{-\frac{1}{2}}\beta'\|G\|_2 + (nm)^{-\frac{1}{2}}\beta''\|\mathcal{M}\|_2\big).$$

For comparison, under uniform meta-stability with constant $\beta$, and almost surely bounded loss $0 \leq \ell \leq L$, Wang & Arora (2024) show that with probability at least $1 - \delta$,

$$R(A(\mathbb{S}), \mathbb{S} - R(A(\mathbb{S}), \mu)$$
$$\lesssim \beta \log(mn) \log(1/\delta) + L\sqrt{\log(1/\delta)/(mn)}.$$

To achieve vanishing bounds, their result requires $\beta \ll (\log(mn))^{-1}$, whereas ours requires faster decay $\|H\|_q = O(1), \|G\|_q = O(1), \|\mathcal{M}\|_q = O(1), \beta \ll (mn)^{-1/2}$, $\beta' \ll n^{-1/q}m^{1-1/q}$, and $\beta'' \ll (mn)^{1-1/q}$ for $q = 2, p$.

## 4. Numerical experiments

In this section, we provide some empirical studies highlighting the tightness of our theoretical bounds. Consider i.i.d. observations $S := \{z_i = (x_i, y_i)\}_{i=1}^m \in \mathbb{R}^d \times \mathbb{R}$ from the linear model $Y = X\beta + \varepsilon$, where the errors satisfy $\varepsilon \overset{d}{=} U_1^{-1/\nu} - U_2^{-1/\nu}$ with $U_1, U_2 \overset{i.i.d.}{\sim} U[0,1]$. Note that here $p = \nu/2$. We set $d = 5$, $\beta = (1, 1, \ldots, 1)^\top$, and vary $m \in \{500, 1000\}$, and $\nu = \{2.2, 4.4\}$. If the bound in Theorem 3.4 is sharp, the ratio $p(y) := \frac{\mathbb{P}(|R - R_{emp}| > y)}{\mathbb{P}(|R - R_{emp}| > C_0 y)}$ should stabilize near $C_0^{\nu/2}$ for large $y$. We set $C_0 = 1.5$. To incorporate stability in our analysis, we use ridge regression for empirical risk minimization with regularization parameter $\lambda = 1.0$. Tail probabilities are empirically estimated via $50,000$ Monte Carlo draws. Figure 1 shows that $p(y)$

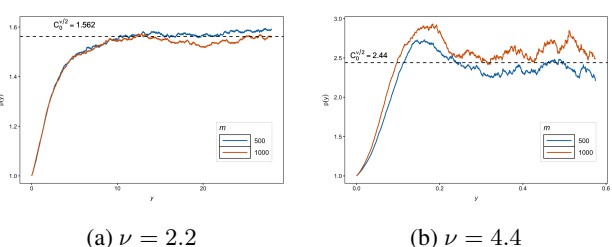

(a) $\nu = 2.2$         (b) $\nu = 4.4$

*Figure 1.* Plot of $p(y)$ versus $y$; both curves stabilize around $C_0^{\nu/2}$ for large $y$.

exhibits initial exponential growth before slowing down and stabilizing near $C_0^{\nu/2}$, further vindicating the importance of

the polynomial-in-$y$ term in Theorem 3.4. For larger $\nu$ (e.g., $\nu = 4.4$), the Gaussian tail dominates the polynomial tail more strongly at smaller values of $y$. Consequently, $p(y)$ may exceed the threshold $C_0^{\nu/2}$ initially, before stabilizing in the large-$y$ regime. This behavior confirms the tightness of our results.

Moreover, as baby steps towards more practically relevant stability analysis, we perform similar experiments on two layer neural-network, trained via ADAM (Kingma & Ba, 2015), for a regression problem. The corresponding plot of $p(y)$ versus $y$ appears in Figure 2. In this result, even for a modern training algorithm, the heavy-tailed nature of the data shines through in distinct characterizations of Gaussian and polynomial tails. Additional details and experiments for transductive learning appear in Appendix § D.

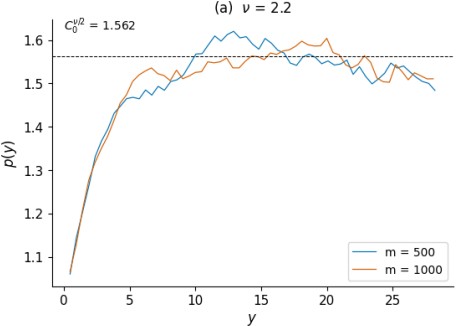

*Figure 2.* Plot of $p(y)$ versus $y$ for the neural network regression experiment in Section D.3.

## 5. Conclusion and Future Works

In this work, we provide a systematic treatment of stability analysis under weakened $L_p$ assumptions by establishing a new large-deviation inequality, which may be of independent interest. Theorems 2.2 and 2.6 broaden the scope of stability-based generalization by accommodating settings where the stability increment admits either higher-order finite moments ($p \geq 2$) or only lower-order moments ($p \in (1, 2)$). We develop three representative applications in Section 3, and provide supporting empirical evidence for the sharpness of our bounds in Section 4.

Our results also suggest several natural directions for future research: **(i)** extending the analysis to dependent data (e.g., Markovian sampling) (Yang et al., 2021; Wang et al., 2022); **(ii)** extending the framework to unsupervised learning problems (Cheng et al., 2021); and **(iii)** establishing matching lower bounds in specific settings to formally quantify when the polynomial correction term is unavoidable under finite-moment assumptions.

## Impact Statement

This paper develops theoretical tools for stability-based generalization under unbounded losses, replacing classical boundedness or sub-Weibull assumptions with finite $L_p$ moment conditions on one-sample perturbations. By enabling high-probability generalization guarantees in settings where exponential-tail assumptions are implausible, the results may improve the theoretical foundations of learning with heavy-tailed data (e.g., robust regression, learning under contamination, and other non-ideal data regimes). This can support more reliable model assessment and encourage principled handling of outliers and unbounded losses.

The work is primarily theoretical and does not introduce new learning systems, datasets, or deployment-facing algorithms. As with most generalization theory, the results could be misinterpreted as a blanket endorsement of a model's safety or suitability for high-stakes deployment. Our bounds certify statistical generalization under specific stability and moment conditions; they do not address fairness, privacy, security, or downstream harms. Improperly claiming "theory-backed reliability" without validating the assumptions (e.g., finite moments or sufficient stability decay) could create false confidence.

We do not foresee direct societal harms arising from this theoretical contribution. We encourage practitioners who use these insights to (i) empirically validate the relevant stability/moment conditions when possible, and (ii) complement generalization guarantees with domain-appropriate safety, robustness, and ethical evaluations.

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

# Appendices

In this appendix, we provide additional discussion of the technical novelty, the proofs of our technical results, and additional experimental evidence behind the tightness of our theoretical results. In particular, §A connects our work to two closely related papers in the literature. §B collects the proofs of our main technical results Theorem 2.2 and 2.6. §C.1, §C.2 and §C.3 contains the proofs of results in §3.1, §3.2 and §3.3 respectively. Finally, we provide some additional simulation results in §D.

## A. Technical novelty compared to existing literature

In this section, we provide additional discussion connecting this manuscript to Celisse & Guedj (2016) and Li & Liu (2024).

The main contribution of Celisse & Guedj (2016) is to introduce $L_q$-stability as a higher-moment stability notion for the leave-one-out (LOO) estimator and to show, for concrete models such as ridge regression, how this condition can be verified and then combined with moment inequalities to obtain PAC-exponential LOO bounds. Our contribution differs in emphasis: starting from a replace-one perturbation bound dominated by a random envelope $H$ with only a fixed finite $L_q$ moment, we derive a high-probability Nagaev-type concentration and generalization bound. Thus, Celisse & Guedj (2016) primarily studies when higher-moment stability holds, whereas our paper studies what high-probability guarantees follow from a finite-moment replace-one stability condition. Their work is best viewed as closely related model-side evidence that higher-moment stability conditions are meaningful, complementing the motivation for ours.

The work of Li & Liu (2024) is structurally closer to our framework, particularly Theorems 4 and 5 and the subsequent Remark 6. Theorem 5 therein provides a symmetrization-like argument that may be leveraged to deduce exponential tails even under polynomial moment assumptions. In contrast, our proof strategy hinges more on a clever truncation-trick, borrowing on ideas from Nagaev (1979). On the other hand, Li & Liu (2024), in their Corollary 29, also attempts to derive a similar Gaussian-plus-polynomial deviation bound by leveraging Theorems 4 and 5 therein. Unfortunately, it can be seen that the proof contains a fatal gap that cannot be trivially rescued. In the polynomial-moment regime, Corollary 28 provides only a fixed-$p$ moment bound, since the assumption is merely that there exists some $p \geq 2$ such that $\mathbb{E}|f_i(X)(x)|^p \leq b_i$. The proof of Corollary 29 then applies Markov's inequality to obtain a $t^{-p}$ tail, and in the variance-dominated case chooses a special threshold $t \asymp \sqrt{p}(\sum_i \sigma_i^2)^{1/2}$ so that this polynomial bound equals $e^{-p}$. This only gives a bound at that particular threshold. The subsequent step, where this is rewritten as a sub-Gaussian bound for arbitrary $t > 0$, is not justified unless one can vary $p$ with $t$, i.e. unless a whole family of moment bounds is available. That is not the stated assumption of Corollary 28. Hence, while Corollary 29 has essentially the same formal statement as our Theorem 2.2, its displayed proof does not justify the full Gaussian-plus-polynomial inequality from the stated polynomial-moment assumption. In principle, a large deviation bound similar to our Theorem 2.2 may also be developed using clever chaining arguments on top of Theorem 5 of Li & Liu (2024), but it is not an obvious extension. In contrast, our Theorem 2.2 goes through a different route to provide a direct and self-contained derivation of this Nagaev-type bound in the stability setting, settling the problem for once and all.

## B. Proofs of Theorems in Section 2

### B.1. Technical lemmas

**Lemma B.1.** *Let $X \in \mathcal{X}$ be a random variable and a function $g : \mathcal{X} \to \mathbb{R}$ satisfying $\mathbb{P}(g(X) > b) = 0, b > 0$, and for any $p \geq 2$*

$$\|g(X)\|_p < \infty. \tag{12}$$

*Then for any positive $\phi$,*

$$\log \mathbb{E} \exp\{\phi g(X)\} \leq \phi \mathbb{E} g(X) + \frac{e^p \phi^2}{2} \mathbb{E}[g(X)^2] + \frac{e^{\phi b} - 1 - \phi b}{b^p} \mathbb{E}[|g(X)|^p] \times \mathbb{I}\{\phi > p/b\}. \tag{13}$$

**Proof of Lemma B.1.** By Taylor expansion,

$$\mathbb{E} \exp\{\phi g(X)\} = 1 + \phi \mathbb{E} g(X) + \int_{g(X) \leq b} (e^{\phi g(X)} - 1 - \phi g(X)) \mathrm{d}F(g(X)). \tag{14}$$

Consider 2nd-order Taylor formula with integral remainder,

$$e^x = 1 + x + x^2 \int_0^1 (1-s)e^{sx}\mathrm{d}s. \tag{15}$$

Then for any $x \leq t$, when $s \in [0,1]$, $sx \leq x \leq t$, we have

$$e^x - 1 - x = x^2 \int_0^1 (1-s)e^{sx}\mathrm{d}s \leq x^2 \int_0^1 (1-s)e^t\mathrm{d}s = \frac{e^t x^2}{2}. \tag{16}$$

Therefore, suppose that $\phi \leq p/b$, then

$$\int_{g(X)\leq b} (e^{\phi g(X)} - 1 - \phi g(X))\mathrm{d}F(g(X)) \leq \int_{g(X)\leq p/\phi} (e^{\phi g(X)} - 1 - \phi g(X))\mathrm{d}F(g(X)) \leq \frac{e^p \phi^2}{2}\mathbb{E}[g(X)^2]. \tag{17}$$

Then, for $\phi > p/b$,

$$\int_{g(X)\leq b} (e^{\phi g(X)} - 1 - \phi g(X))\mathrm{d}F(g(X))$$
$$\leq \int_{g(X)\leq p/\phi} (e^{\phi g(X)} - 1 - \phi g(X))\mathrm{d}F(g(X)) + \int_{g(X)>p/\phi} \frac{e^{\phi g(X)} - 1 - \phi g(X)}{g(X)^p} g(X)^p \mathrm{d}F(g(X)). \tag{18}$$

Since $\frac{e^{\phi g(X)} - 1 - \phi g(X)}{g(X)^p}$ increase w.r.t. $g(X)$ for $g(X) > p/\phi$, on the region $p/\phi < g(X) \leq b$ we have $g(X) > 0$. thus

$$\int_{g(X)>p/\phi} \frac{e^{\phi g(X)} - 1 - \phi g(X)}{g(X)^p} g(X)^p \mathrm{d}F(g(X))$$
$$\leq \frac{e^{\phi b} - 1 - \phi b}{b^p} \int_{g(X)>p/\phi} |g(X)|^p \mathrm{d}F(g(X)) \leq \frac{e^{\phi b} - 1 - \phi b}{b^p}\mathbb{E}[|g(X)|^p]. \tag{19}$$

From Equation (17) and Equation (18), with $\log x \leq x - 1$, we obtain the bound in Equation (13). $\qquad \square$

## B.2. Proof of Theorem 2.2

**Proof of Theorem 2.2.** Let $\mathcal{F}_k = (\ldots, X_{k-1}, X_k)$ and define the projection operator $\mathcal{P}_i(\cdot) = \mathbb{E}(\cdot|\mathcal{F}_i) - \mathbb{E}(\cdot|\mathcal{F}_{i-1})$. For simplicity of notation, denote $\Delta_n = f(X_1, X_2, \ldots, X_n) - \mathbb{E}f(X_1, X_2, \ldots, X_n)$. Then, we can obtain the following decomposition,

$$|\Delta_n| = |\sum_{i\leq n} \mathcal{P}_i(g)| = |\sum_{i\leq n}\mathbb{E}(g - g_i|\mathcal{F}_i)|, \tag{20}$$

where $g = f(X_1, X_2, \ldots, X_{i-1}, X_i, X_{i+1}, \ldots, X_n)$ and $g_i = f(X_1, X_2, \ldots, X_{i-1}, X_i', X_{i+1}, \ldots, X_n)$ is the coupled version of $g$ obtained by replacing $X_i$ by an i.i.d. copy $X_i'$.

For any $y = pz/4(p+2)$ and $i \in [n]$, define the truncated version of $\tilde{\mathcal{P}}_i(g) = \mathcal{P}_i(g)\mathbf{1}\{|\mathcal{P}_i(g)| \leq y\}$. Note that if every increment satisfies $|\mathcal{P}_i(\Delta_n)| < y$, then $\mathcal{P}_i(\Delta_n) = \tilde{\mathcal{P}}_i(\Delta_n)$ for all i, so

$$\Delta_n = \sum_{i=1}^n \mathcal{P}_i(\Delta_n) = \sum_{i=1}^n \tilde{\mathcal{P}}_i(\Delta_n). \tag{21}$$

Hence, on the event $\{|\Delta_n| > z\}$, we have either the truncated sum itself exceeds $z$, or at least one increment satisfies $|\mathcal{P}_i(\Delta_n) \geq y|$. Consequently, we can obtain the following decomposition

$$\mathbb{P}(|\Delta_n| > z) \leq \mathbb{P}(|\sum_{i\leq n}\tilde{\mathcal{P}}_i(\Delta_n)| \geq z) + \sum_{i=1}^n \mathbb{P}(|\mathcal{P}_i(\Delta_n)| \geq y) =: I_1 + I_2. \tag{22}$$

For $I_2$ in (22), by Markov inequality, one derives

$$I_2 \leq \frac{\sum_{i=1}^n \mathbb{E}|\mathcal{P}_i(g)|^p}{y^p} \leq \frac{\sum_{i=1}^n \mathbb{E}|H_i(X_i, X_i')|^p}{y^p}, \tag{23}$$

where the second inequality comes from

$$\mathbb{E}|\mathcal{P}_i(g)|^p = \mathbb{E}|\mathbb{E}(g - g_i|\mathcal{F}_i)|^p \leq \mathbb{E}[\mathbb{E}(|g - g_i|^p|\mathcal{F}_i)] = \mathbb{E}|g - g_i|^p \leq \mathbb{E}|H_i(X_i, X_i')|^p. \tag{24}$$

Then, we bound the term $I_1$. Define

$$\mu_i := \mathbb{E}[\tilde{\mathcal{P}}_i(g)|\mathcal{F}_{i-1}], \qquad U_i := \tilde{\mathcal{P}}_i(g) - \mu_i, \tag{25}$$

where

$$\begin{align}
|\mu_i| =&|\mathbb{E}[\tilde{\mathcal{P}}_i|\mathcal{F}_{i-1}]| = |\mathbb{E}[\mathcal{P}_i(g)\mathbf{1}\{|\mathcal{P}_i(g)| \leq y\}|\mathcal{F}_{i-1}]| \tag{26} \\
=&|-\mathbb{E}[\mathcal{P}_i(g)\mathbf{1}\{|\mathcal{P}_i(g)| > y\}|\mathcal{F}_{i-1}]| \tag{27} \\
\leq&\mathbb{E}[|\mathcal{P}_i(g)|\mathbf{1}\{|\mathcal{P}_i(g)| > y\}|\mathcal{F}_{i-1}] \tag{28} \\
\leq&y^{1-p}\mathbb{E}[|\mathcal{P}_i(g)|^p|\mathcal{F}_{i-1}] \tag{29} \\
\leq&y^{1-p}\mathbb{E}|H_i(x_i, x_i')|^p. \tag{30}
\end{align}$$

Thus we have,

$$\mathbb{E}[U_i|\mathcal{F}_{i-1}] = 0, \qquad |U_i| \leq 2y \tag{31}$$

and term $I_1$ satisfies,

$$I_1 \leq \mathbb{P}(|\sum_{i=1}^n U_i| \geq z/2) + \mathbf{1}\left\{\sum_{i=1}^n |\mu_i| > z/2\right\} \leq P(|\sum_{i=1}^n U_i| \geq z/2) + \frac{2\sum_{i=1}^n \mathbb{E}|H_i(x_i, x_i')|^p}{zy^{p-1}} \tag{32}$$

For the term $\mathbb{P}(|\sum_{i=1}^n U_i| \geq z/2)$, we need to control the following moment generating function

$$M(t) = \mathbb{E}\exp\left\{t\sum_{h=1}^n U_i\right\}, \, t > 0. \tag{33}$$

To that end, observe that for each $h \in [n]$, $|U_i| \leq 2y$, by Lemma B.1, we have

$$\begin{align}
\mathbb{E}\{\exp(tU_i)|\mathcal{F}_{i-1}\} &\leq 1 + \frac{e^p t^2}{2}\mathbb{E}[|U_i|^2|\mathcal{F}_{i-1}] + \frac{\exp(2ty) - 1 - 2ty}{(2y)^p}\mathbb{E}[|U_i|^p|\mathcal{F}_{i-1}] \times 1\{t > p/2y\} \\
&\leq 1 + \frac{e^p t^2}{2}\mathbb{E}|H_i(X_i, X_i')|^2 + \frac{\exp(2ty) - 1 - 2ty}{y^p}\mathbb{E}|H_i(X_i, X_i')|^p \times 1\{t > p/2y\}, \tag{34}
\end{align}$$

where the upper bound above is independent of $\mathcal{F}_{i-1}$. Therefore, iterated from 1 to $n$, we can obtain that

$$\begin{align}
M(t) &\leq \prod_{i=1}^n \left(1 + \frac{e^p t^2}{2}\mathbb{E}|H_i(X_i, X_i')|^2 + \frac{\exp(2ty) - 1 - 2ty}{y^p}\mathbb{E}|H_i(X_i, X_i')|^p \times 1\{t > p/2y\}\right) \\
&\leq \exp\left(\frac{1}{2}e^p t^2 \sum_{i=1}^n \mathbb{E}|H_i(X_i, X_i')|^2 + \frac{\exp(2ty) - 1 - 2ty}{y^p}\sum_{i=1}^n \mathbb{E}|H_i(X_i, X_i')|^p \times 1\{t > p/2y\}\right). \tag{35}
\end{align}$$

By Chernoff's bound, it is obvious that

$$\mathbb{P}(|\Delta_n| > z) \leq \frac{\sum_{i=1}^n \mathbb{E}|H_i(X_i, X_i')|^p}{y^p} + \frac{2\sum_{i=1}^n \mathbb{E}|H_i(X_i, X_i')|^p}{zy^{p-1}}$$

$$+ 2\exp\left(-tz/2 + \frac{1}{2}e^p t^2 \sum_{i=1}^n \mathbb{E}|H_i(X_i, X_i')|^2 + \frac{\exp(2ty) - 1 - 2ty}{y^p}\sum_{i=1}^n \mathbb{E}|H_i(X_i, X_i')|^p \times 1\{t > p/2y\}\right)$$

$$:= \frac{\sum_{i=1}^n \mathbb{E}|H_i(X_i, X_i')|^p}{y^p} + \frac{2\sum_{i=1}^n \mathbb{E}|H_i(X_i, X_i')|^p}{zy^{p-1}} + 2\mathcal{I}_3. \tag{36}$$

Then, for simplicity of representation, denote $M_2 = \sum_{i=1}^n \mathbb{E}|H_i(X_i, X_i')|^2$, $M_p = \sum_{i=1}^n \mathbb{E}|H_i(X_i, X_i')|^p$. Define

$$h(t) = \frac{e^p t^2}{2}M_2 - t\frac{z}{2}, \tag{37}$$

$$h_1(t) = \frac{e^p t^2}{2}M_2 - \alpha t\frac{z}{2}, \quad 0 < \alpha < 1, \tag{38}$$

$$h_2(t) = \frac{\exp(2ty) - 1 - 2ty}{y^p}M_p - \gamma t\frac{z}{2}, \quad \gamma = 1 - \alpha. \tag{39}$$

By taking derivative of $h_1(t)$ and $h_2(t)$ w.r.t. $t$, then we can get

$$t_1 = \alpha z/(2e^p M_2), \quad t_2 = \frac{1}{2y}\log(\gamma zy^{p-1}/4M_p + 1), \tag{40}$$

where $h_1'(t_1) = 0$ and $h_2'(t_2) = 0$, which implies that $h_1(t)$ and $h_2(t)$ reaches their minimum at $t_1$ and $t_2$ separately. First we examine the case when $t_1 \leq p/2y$, we can conclude that

$$\mathcal{I}_3 \leq \exp\left\{-\frac{\alpha^2 z^2}{8e^p M_2}\right\}. \tag{41}$$

Then, for $t_2 > t_1 \geq p/2y$, plug-in $t = t_1$, $\mathcal{I}_3$ becomes

$$\mathcal{I}_3 \leq \exp\{h_1(t_1) + h_2(t_1)\} \leq \exp\{h_1(t_1)\} = \exp\left\{-\frac{\alpha^2 z^2}{8e^p M_2}\right\}, \tag{42}$$

where the second inequality comes from $h_2(t_2) < h_2(t_1) < h_2(0) < 0$ since $h_2$ is convex and $t_2$ is the minimizer of $h_2$. Consider when $t_1 > t_2 > p/2y$

$$\begin{aligned}
h_1(t_2) + h_2(t_2) &< t_2\left(\frac{e^p M_2 t_1}{2} - \frac{z}{2}\right) + \frac{(e^{2t_2 y} - 1)M_p}{y^p} \\
&= \frac{\gamma z}{4y} - t_2(1 - \alpha/2)\frac{z}{2} \\
&= \frac{\gamma z}{4y} - \frac{\alpha z t_2}{4} - \gamma t_2\frac{z}{2} \\
&< \left(\gamma - \frac{p\alpha}{2}\right)\frac{z}{4y} - \gamma t_2\frac{z}{2},
\end{aligned} \tag{43}$$

which lead to

$$\mathcal{I}_3 \leq \exp\left\{\left(\gamma - \frac{p\alpha}{2}\right)\frac{z}{4y} - \gamma\frac{z}{4y}\log\left(\frac{\gamma zy^{p-1}}{4M_p} + 1\right)\right\}. \tag{44}$$

It now remains to examine when $t_1 > p/2y \geq t_2$, we only need to consider the case when plug in $t = p/2y$

$$
\begin{aligned}
h(p/2y) &< \frac{e^p M_2}{2} \frac{p^2}{(2y)^2} - \frac{p}{2y} \frac{z}{2} \\
&< \frac{p}{2y} \left( \frac{e^p M_2}{2} t_1 - \frac{z}{2} \right) \\
&= \frac{p}{2y} \left( \frac{\alpha z}{4} - \frac{z}{2} \right) \\
&= -\gamma \frac{z}{2} \frac{p}{2y} - \frac{\alpha z p}{8y} \\
&< -\gamma \frac{z}{2} t_2 - \frac{\alpha z p}{8y},
\end{aligned}
\tag{45}
$$

thus we get

$$
\mathcal{I}_3 \leq \exp\left\{ \left( \gamma - \frac{p\alpha}{2} \right) \frac{z}{4y} - \gamma \frac{z}{4y} \log\left( \frac{\gamma z y^{p-1}}{4M_p} + 1 \right) \right\}.
\tag{46}
$$

Combining all the result above, since either

$$
\mathcal{I}_3 \leq \exp\left\{ -\frac{\alpha^2 z^2}{8 e^p M_2} \right\},
\tag{47}
$$

or

$$
\mathcal{I}_3 \leq \exp\left\{ \left( \gamma - \frac{p\alpha}{2} \right) \frac{z}{4y} - \gamma \frac{z}{4y} \log\left( \frac{\gamma z y^{p-1}}{4M_p} + 1 \right) \right\},
\tag{48}
$$

holds for all $z > 0$, we have

$$
\begin{aligned}
\mathbb{P}(|\Delta_n| > z) &= \frac{\sum_{i=1}^n \mathbb{E}|H_i(X_i, X_i')|^p}{y^p} + \frac{2 \sum_{i=1}^n \mathbb{E}|H_i(X_i, X_i')|^p}{z y^{p-1}} + 2\mathcal{I}_3 \\
&\leq \frac{\sum_{i=1}^n \mathbb{E}|H_i(X_i, X_i')|^p}{y^p} + \frac{2 \sum_{i=1}^n \mathbb{E}|H_i(X_i, X_i')|^p}{z y^{p-1}}
\end{aligned}
\tag{49}
$$

$$
+ 2 \exp\left\{ -\frac{\alpha^2 z^2}{8 e^p M_2} \right\} + 2 \exp\left\{ \left( \gamma - \frac{p\alpha}{2} \right) \frac{z}{4y} - \gamma \frac{z}{4y} \log\left( \frac{\gamma z y^{p-1}}{4M_p} + 1 \right) \right\}.
\tag{50}
$$

By taking $\gamma = \frac{p\alpha}{2}$, above equation can be generalized to

$$
\begin{aligned}
\mathbb{P}(|\Delta_n| > z) &\leq \frac{\sum_{i=1}^n \mathbb{E}|H_i(X_i, X_i')|^p}{y^p} + \frac{2 \sum_{i=1}^n \mathbb{E}|H_i(X_i, X_i')|^p}{z y^{p-1}} \\
\end{aligned}
\tag{51}
$$

$$
+ 2 \exp\left\{ -\frac{\alpha^2 z^2}{8 e^p M_2} \right\} + 2 \exp\left\{ \left( \gamma - \frac{p\alpha}{2} \right) \frac{z}{4y} - \gamma \frac{z}{4y} \log\left( \frac{\gamma z y^{p-1}}{4M_p} + 1 \right) \right\}
$$

$$
\leq \frac{\sum_{i=1}^n \mathbb{E}|H_i(X_i, X_i')|^p}{y^p} + \frac{2 \sum_{i=1}^n \mathbb{E}|H_i(X_i, X_i')|^p}{z y^{p-1}}
\tag{52}
$$

$$
+ 2 \exp\left\{ -\frac{\alpha^2 z^2}{8 e^p M_2} \right\} + 2 \left( \frac{\gamma z y^{p-1}}{4M_p} \right)^{-\gamma z/4y}.
\tag{53}
$$

Furthermore, taking $y = \gamma z/4$, the proof is completed. $\qquad\square$

### B.3. Proof of Theorem 2.6

**Proof of Theorem 2.6.** We follow the same martingale–difference decomposition as in the proof of Theorem 2.2. Let

$$
\mathcal{F}_k = \sigma(X_1, \dots, X_k), \qquad \mathcal{P}_i(g) = \mathbb{E}(g \mid \mathcal{F}_i) - \mathbb{E}(g \mid \mathcal{F}_{i-1}),
\tag{54}
$$

and write

$$\Delta_n := f(X_1, \ldots, X_n) - \mathbb{E}f(X_1, \ldots, X_n) = \sum_{i=1}^{n} \mathcal{P}_i(g). \tag{55}$$

For a threshold $y > 0$ (to be chosen in terms of $z$ and $p$ below) set

$$\widetilde{\mathcal{P}}_i(g) := \mathcal{P}_i(g)\mathbf{1}\{|\mathcal{P}_i(g)| \leq y\}. \tag{56}$$

Define

$$\mu_i = \mathbb{E}[\mathcal{P}_i(g)\mathbf{1}\{|\mathcal{P}_i(g)| \leq y\}|\mathcal{F}_{i-1}], \qquad U_i = \mathcal{P}_i(g)\mathbf{1}\{|\mathcal{P}_i(g)| \leq y\} - \mu_i, \tag{57}$$

thus we have,

$$\mathbb{E}[U_i|\mathcal{F}_{i-1}] = 0, \qquad |U_i| \leq 2y, \tag{58}$$

similarly as what we have done in the proof of Theorem 2.2,

$$|\mu_i| \leq y^{1-p}\mathbb{E}|H_i(x_i, x_i')|^p \tag{59}$$

Then, we have the following decomposition

$$\mathbb{P}(|\Delta_n| > z) \leq \mathbb{P}\Big(|\sum_{i=1}^{n} U_i| \geq z/2\Big) + \mathbf{1}\Big\{\sum_{i=1}^{n} |\mu_i| > z/2\Big\} + \sum_{i=1}^{n} \mathbb{P}(|\mathcal{P}_i(g)| > y) =: I_1 + I_2 + I_3. \tag{60}$$

By Lipschitz property, we have

$$\mathbb{P}\big(|\mathcal{P}_i(g)| > y\big) \leq \mathbb{P}(|H_i(X_i, X_i')| > y). \tag{61}$$

Write

$$S_n := \sum_{i=1}^{n} U_i. \tag{62}$$

Each $U_i$ is $\mathcal{F}_i$–measurable, centered ($\mathbb{E}[U_i \mid \mathcal{F}_{i-1}] = 0$) and bounded by $2y$. Then, defining $u := \widetilde{\mathcal{P}}_i(g)$, we can derive the following

$$\begin{aligned} \mathbb{E}[\exp(tu)|\mathcal{F}_{i-1}] &\leq 1 + \int_{|u| \leq 2y} \frac{e^{tu} - 1 - tu}{u^2} u^2 \mathrm{d}F(u|\mathcal{F}_{i-1}) \\ &\leq 1 + \frac{e^{2ty} - 1 - 2ty}{(2y)^2} \int_{|u| \leq 2y} u^2 \mathrm{d}F(u|\mathcal{F}_{i-1}) \\ &\leq 1 + \frac{e^{2ty} - 1 - 2ty}{(2y)^p} \int_{|u| \leq 2y} |u|^p \mathrm{d}F(u|\mathcal{F}_{i-1}), \end{aligned} \tag{63}$$

where the second inequality comes from the monotonicity of $\frac{e^{tu}-1-tu}{u^2}$ w.r.t. $u \leq 2y$ and the third inequality comes from monotonicity of $\frac{|u|^p}{y^p}$ w.r.t. $p$ for $u > 0$.

By $\log x \leq x - 1$, above result can be expressed as

$$\log \mathbb{E}\exp(U_i) \leq \frac{e^{2ty} - 1 - 2ty}{(2y)^p}\mathbb{E}[|\widetilde{\mathcal{P}}_i(g)|^p|\mathcal{F}_{i-1}] \leq \frac{e^{2ty} - 1 - 2ty}{y^p}\mathbb{E}|H_i(x_i, x_i')|^p \tag{64}$$

Taking the sum over $i$, we can further obtain

$$e^{-tz/2}\mathbb{E}\exp(\sum_{i=1}^{n} U_i) \leq \exp\left\{\frac{e^{2ty} - 1 - 2ty}{y^p}\sum_{i=1}^{n}\mathbb{E}|H_i(x_i, x_i')|^p - t\frac{z}{2}\right\}. \tag{65}$$

Setting

$$t = \frac{1}{2y} \log \left\{ \frac{zy^{p-1}}{4 \sum_{i=1}^{n} \mathbb{E}|H_i(x_i, x_i')|^p} + 1 \right\}, \tag{66}$$

the right hand side becomes

$$
\begin{aligned}
e^{-tz} \mathbb{E} \exp(U_i) &\leq \exp \left\{ \frac{z}{4y} - \left( \frac{z}{4y} + \frac{\sum_{i=1}^{n} \mathbb{E}|H_i(x_i, x_i')|^p}{y^p} \right) \log \left( \frac{zy^{p-1}}{4 \sum_{i=1}^{n} \mathbb{E}|H_i(x_i, x_i')|^p} + 1 \right) \right\} \\
&\leq \exp \left\{ \frac{z}{4y} \left( 1 - \log \left( \frac{zy^{p-1}}{4 \sum_{i=1}^{n} \mathbb{E}|H_i(x_i, x_i')|^p} \right) \right) \right\} \\
&= \left( \frac{4e \sum_{i=1}^{n} \mathbb{E}|H_i(x_i, x_i')|^p}{zy^{p-1}} \right)^{z/4y}. \tag{67}
\end{aligned}
$$

Combining the bounds for $I_1$ and $I_2$ and plug in $y = \frac{z}{4Q}$ for any $Q > 1$, the proof is completed.

For the term indicator term $\mathbf{1} \left\{ \sum_{i=1}^{n} |\mu_i| > z/2 \right\}$, consider two case separately. Let $M_p = \sum_{i=1}^{n} \mathbb{E}|H_i(x_i, x_i')|^p$, define $B_y := \sum_{i=1}^{n} |\mu_i| \leq M_p/y^{p-1}$. If $B_y \geq z/2$, we have $M_p > \frac{zy^{p-1}}{2} = z^p/(2(4Q)^{p-1})$, thus $A = e4^p Q^{p-1} M_p/z^p > 2e$. Indicator term can be adsorbed by $I_1$. Otherwise, Indicator term is 0. $\qquad\square$

# C. Proofs of Theorems in Section 3

## C.1. Proof of Results in Section 3.1: Empirical Risk Minimization

**Proof of Theorem 3.4.** Clearly,

$$|R - R^i| \le |\mathbb{E}_z[\ell(A_S, z) - \ell(A_{S^i}, z)]| \le \beta H(z_i, z'_i). \tag{68}$$

On the other hand,

$$|R_{loo} - R^i_{loo}| \le \frac{1}{m}\sum_{j \neq i}|\ell(A_{S \setminus j}, z_j) - \ell(A_{S^i, \setminus j}, z_j)| + \frac{1}{m}|\ell(A_{S \setminus i}, z_i) - \ell(A_{S \setminus i}, z'_i)| \le \frac{m-1}{m}\beta H(z_i, z'_i) + \frac{1}{m}\beta' G(z_i, z'_i). \tag{69}$$

Therefore, with $\phi = R - R_{loo}$ and $\phi^i = R^i - R^i_{loo}$, (68) and (69) yields

$$|\phi - \phi^i| \le \frac{2m-1}{m}\beta H(z_i, z'_i) + \frac{1}{m}\beta' G(z_i, z'_i).$$

Moreover,

$$\mathbb{E}_S[R - R_{loo}] = R_m - \frac{1}{m}\sum_{j=1}^m \mathbb{E}_S[\ell(A_{S \setminus j}, z_j)] = R_m - R_{m-1}.$$

Applying Theorem 2.2, we can get

$$\mathbb{P}(|R - R_{loo} - (R_m - R_{m-1})| > y) \le c_1 \frac{m\mathbb{E}|\frac{2m-1}{m}\beta H(z, z') + \beta'\frac{G(z,z')}{m}|^p}{y^p} + 2\exp\{-c_2\frac{y^2}{m\mathbb{E}|\frac{2m-1}{m}\beta H(z,z') + \beta'\frac{G(z,z')}{m}|^2}\}. \tag{70}$$

Note that by Hölder's inequality, we can have for $q \in \{2, p\}$

$$\mathbb{E}\left|\frac{2m-1}{m}\beta H(z_i) + \frac{G}{m}\right|^q \le 2^q \mathbb{E}\left[\left(\frac{2m-1}{m}\beta\right)^q|H(z_i)|^q + \frac{1}{m^q}|\beta'G|^q\right]$$

$$\le 2^q\left((2\beta)^q\mathbb{E}|H(z_i)|^q + \frac{(\beta')^q}{m^q}\mathbb{E}|G|^q\right),$$

thus taking sum over $1$ to $m$, we can obtain Equation (5).

For the $R_{emp}$, we proceed similarly. We have

$$|R_{emp} - R^i_{emp}| \le \frac{1}{m}\sum_{j \neq i}|\ell(A_S, z_j) - \ell(A_{S^i}, z_j)| + \frac{1}{m}|\ell(A_S, z_i) - \ell(A_{S^i}, z_i)| + \frac{1}{m}|\ell(A_{S^i}, z_i) - \ell(A_{S^i}, z'_i)|$$

$$\le \beta H(z_i, z'_i) + \beta'\frac{G(z_i, z'_i)}{m} \tag{71}$$

and

$$\mathbb{E}_S[R - R_{emp}] = \mathbb{E}_{S, z'_i}[\ell(A_S, z'_i)] - \frac{1}{m}\sum_{j=1}^m \mathbb{E}_S[\ell(A_S, z_j)]$$

$$\le \frac{1}{m}\sum_{j=1}^m (\mathbb{E}_{S, z'_i}[\ell(A_{S^j}, z_j)] - \mathbb{E}_{S, z'_i}[\ell(A_S, z_j)]) \quad \text{(where } S^j = (z_1, \ldots, z_{j-1}, z'_i, z_{j+1, \ldots, z_n}))$$

$$\le \beta\mathbb{E}[H(z, z')]. \tag{72}$$

So the Theorem 2.2 can again be applied and the proof is completed.

$\square$

## C.2. Proof of Results in Section 3.2: Transductive Regression Algorithms

**Proof of Theorem 3.9.** Let $\mathcal{F}_i := \sigma(x_1, \ldots, x_i)$ and define the Doob martingale

$$M_i := \mathbb{E}[\varphi \mid \mathcal{F}_i], \qquad i = 0, 1, \ldots, m, \tag{73}$$

so that $M_0 = \mathbb{E}\varphi$ and $M_m = \varphi$. Let the martingale differences be

$$D_i := M_i - M_{i-1}, \qquad i = 1, \ldots, m, \tag{74}$$

so $\varphi - \mathbb{E}\varphi = \sum_{i=1}^m D_i$ and $\mathbb{E}[D_i \mid \mathcal{F}_{i-1}] = 0$.

For fixed $i$, consider $x_i \in X$,

$$G_i(x_i) := \mathbb{E}[\varphi \mid \mathcal{F}_{i-1}, x_i]. \tag{75}$$

Then $M_i = G_i(x_i)$ and $M_{i-1} = \mathbb{E}[G_i(x_i) \mid \mathcal{F}_{i-1}]$, hence

$$D_i = G_i(x_i) - \mathbb{E}[G_i(x_i) \mid \mathcal{F}_{i-1}]. \tag{76}$$

Then, Consider the sequence of $\mathrm{x}_i^m$, since the function $\phi$ is symmetric, any permutations containing same elements may lead to same value. Thus, we only need to consider for the case that $x_i$ is not included in sequence $\mathrm{x}_{i+1}^m$, and the number of those cases are $(m-i)! \binom{N-i-1}{m-i}$

Thus, we have that for all $x_i, x_i' \in R_{i-1}$,

$$|D_i| \leq \frac{u!}{(N-i)!}(m-i)! \binom{N-i-1}{m-i} |\varphi(\mathrm{x}_1^{i-1}, \mathrm{x_i}, \mathrm{x}_{i+1}^m) - \varphi(\mathrm{x}_1^{i-1}, \mathrm{x_i'}, \mathrm{x}_{i+1}^m)| \leq \frac{u}{N-i} H(x_i, x_i') \tag{77}$$

Then, by similar proof as Theorem 2.2, we have

$$\mathbb{P}\left( \sum_{i=1}^m D_i > z \right) \leq 4 \left( 4\frac{p+2}{p} \right)^p \frac{S_p}{z^p} + \exp\left( -\frac{1}{2(p+2)^2 e^p} \frac{z^2}{S_2} \right), \tag{78}$$

where

$$S_p = \mathbb{E}|H(x_i, x_i')|_p^p \sum_{i=1}^m \left( \frac{u}{N-i} \right)^p. \tag{79}$$

Then, consider for $p = 2$,

$$\sum_{i=1}^m \left( \frac{u}{N-i} \right)^2 \leq \frac{mu}{(N-1/2)(1-1/2\max\{m,u\})}, \tag{80}$$

for $p > 2$,

$$\sum_{i=1}^m \left( \frac{u}{m+n-i} \right)^p \leq \frac{u^p}{p-1} \left( \frac{1}{(u-1/2)^{p-1}} - \frac{1}{(m+u-1/2)^{p-1}} \right) \tag{81}$$

The proof is complete.

$\square$

Before concluding the proof of Theorem 3.10, we bound $|\mathbb{E}\phi(S)|$ and the one-swap increment $|\phi(S) - \phi(S')|$, where $S$ and $S'$ differ in exactly one point.

**Lemma C.1.** *Let $\mathcal{H}$ be an $L_p$-bounded hypothesis class and let $A$ be an $L_p$-stable algorithm. Let $h$ and $h'$ be the hypotheses returned by $A$ when trained on $S = \{x_1, \ldots, x_i, \ldots, x_m\}$ and $S' = \{x_1, \ldots, x_i', \ldots, x_m\}$, which differ in exactly one point, where $x_i'$ is an independent copy of $x_i$ for any $i \in [1, m]$. Define $\phi(S) = R(h) - \hat{R}(h)$. Then, for any $p \geq 2$,*

$$|\phi(S) - \phi(S')| \leq 2\beta H(x, x') + \left| \frac{1}{u} - \frac{1}{m} \right| \beta' G(x, x').$$

**Lemma C.2.** *Let h be the hypothesis returned by an $L_p$-stable algorithm A trained on $S = \{x_1, \ldots, x_m\}$. Then,*

$$\left|\mathbb{E}_S[\phi(S)]\right| \leq \frac{\beta}{m} \sum_{i=1}^{m} \mathbb{E}H(x_i, x_i').$$

**Proof of Lemma C.1.** WLOG, we assume $x_i' = x_{m+j}$ for $j \in [1, u]$. By definition and assumption, we have

$$
\begin{aligned}
\left|\phi(S) - \phi(S')\right| = &\Big| \frac{1}{u} \sum_{k \in [1,u], k \neq j} \Big( \ell(h(S), x_{m+k}) - \ell(h(S'), x_{m+k}) \Big) - \frac{1}{m} \sum_{\ell \in [1,m], \ell \neq i} \Big( \ell(h(S), x_\ell) - \ell(h(S'), x_\ell) \Big) \\
&+ \frac{1}{u} \Big( \ell(h(S), x_{m+j}) - \ell(h(S'), x_i) \Big) - \frac{1}{m} \Big( \ell(h(S), x_i) - \ell(h(S'), x_{m+j}) \Big) \Big| \\
\leq &\frac{u-1}{u} H(x, x') + \frac{m-1}{m} H(x, x') \\
&+ \min\{\frac{1}{u}, \frac{1}{m}\} \Big( \left| \ell(h(S), x_{m+j}) - \ell(h(S'), x_{m+j}) \right| + \left| \ell(h(S), x_i) - \ell(h(S'), x) \right| \Big) \\
&+ \left| \frac{1}{u} - \frac{1}{m} \right| \times \Big( \left| \ell(h(S), x_{m+j}) - \ell(h(S), x_i) \right| + \left| \ell(h(S), x_i) - \ell(h(S'), x_i) \right| \Big) \\
\leq &2\beta H(x, x') + \left| \frac{1}{u} - \frac{1}{m} \right| \beta' G(x, x')
\end{aligned}
\tag{82}
$$

$\square$

**Proof of Lemma C.2.** By definition of $\mathbb{E}[\phi(S)]$,

$$
\begin{aligned}
\mathbb{E}[\phi(S)] &= \mathbb{E}\left[ \frac{1}{u} \sum_{j=1}^{u} \ell(h_S, x_{m+j}) \right] - \mathbb{E}\left[ \frac{1}{m} \sum_{i=1}^{m} \ell(h_S, x_i) \right] \\
&= \frac{1}{u} \sum_{j=1}^{u} \mathbb{E}[\ell(h_S, x_{m+j})] - \frac{1}{m} \sum_{i=1}^{m} \mathbb{E}[\ell(h_S, x_i)] \\
&= \mathbb{E}[\ell(h_S, x_{m+j})] - \mathbb{E}[\ell(h_S, x_i)] \\
&= \mathbb{E}\Big[ \frac{1}{m} \sum_{j=1}^{m} \ell(h_{S^{(j)}}, x_i) \Big] - \mathbb{E}\Big[ \frac{1}{m} \sum_{j=1}^{m} \ell(h_S, x_i) \Big] \\
&= \frac{1}{m} \sum_{j=1}^{m} \mathbb{E}[\ell(h_{S^{(j)}}, x_i) - \ell(h_S, x_i)] \\
&\leq \frac{\beta}{m} \sum_{j=1}^{m} \mathbb{E}H(x_i, x_i')
\end{aligned}
\tag{83}
$$

$\square$

**Proof of Theorem 3.10.** The result follows directly from Theorem 3.9 and Lemmas C.1 and C.2. $\square$

Here, we give a more general result of Transductive regression algorithm. Instead of assuming a single universal bound $H(x_i, x_i')$ as in Assumption 3.6, we allow a weaker, potentially coordinate-dependent control through functions $H_i(x_i, x_i')$.

**Assumption C.3** (Transductive $L_p$-Lipschitz stability). Let $A$ be a transductive learning algorithm. For a partition $\mathcal{X} \vdash (S, T)$, let $h$ be the hypothesis returned by $A$, and let $h'$ be the hypothesis returned for a modified partition $\mathcal{X} \vdash (S', T')$. We say $A$ is uniformly $L_p$-stable with respect to the cost function $\ell$ if there exist nonnegative measurable functions $H_i$ such that, whenever $(S', T')$ is obtained from $(S, T)$ by swapping exactly one point $x_i \in S$ with one point $x_{m+j} \in T$, then for all $x \in \mathcal{X}$,

$$|\ell(h, x) - \ell(h', x)| \leq H_i(x_i, x_{m+j}),$$

and $\|H_i(x_i, x_{m+j})\|_p < \infty$ for some $p \geq 2$.

Under this transductive $L_p$ stability notion, Theorems 3.9 and 3.10 extend to Theorems C.4 and C.5.

**Theorem C.4.** *Let $X$ be a finite set with $|X| = N = m + u$. Let $x_1^m = (x_1, \ldots, x_m)$ be sampled uniformly without replacement from $X$ and let $\phi : X^m \to \mathbb{R}$ be a measurable function. Assume that for each $i \in [m]$ there exists a measurable function $H_i : X \times X \to \mathbb{R}$ such that for all $(x_1, \ldots, x_m) \in X^m$ and all $x_i' \in X$, where $x_i'$ is an independent copy of $x_i$, defining $\varphi = \phi(x_1, \ldots, x_{i-1}, x_i, x_{i+1}, \ldots, x_m)$ and $\varphi_i = \phi(x_1, \ldots, x_{i-1}, x_i', x_{i+1}, \ldots, x_m)$, we have*

$$|\varphi - \varphi_i| \leq H_i(x_i, x_i')$$

*Suppose that for some $p \geq 2$, $\|H_i(x_i, x_i')\|_p < \infty$ for all $i \in [m]$. Then for all $y > 0$,*

$$\mathbb{P}\Big(|\varphi - \mathbb{E}\varphi| > y\Big) \leq c_1 \frac{V_p}{y^p} + 2\exp\Big(-c_2 \frac{y^2}{V_2}\Big), \tag{84}$$

*where for $q \in \{2, p\}$,*

$$V_q := \Big(1 + \frac{m}{N}\Big)^{q-1}\Big(1 + \log\Big(\frac{N}{u}\Big)\Big)\sum_{i=1}^{m}\|H_i(x_i, x_i')\|_q^q, \tag{85}$$

*and $c_1 = 4(4\frac{p+2}{p})^p$ and $c_2 = 1/(2(p+2)^2 e^p)$ are some constants depending only on $p$.*

**Theorem C.5.** *Let $\mathcal{H}$ be a $L_p$-bounded hypothesis class and let $A$ be a transductive $L_p$ stability algorithm satisfying Assumption C.3. Let $h$ be the hypothesis returned by $A$ for a random partition $X \vdash (S, T)$. Then for any $y > 0$, we have*

$$\mathbb{P}\Big(R(h) - \hat{R}(h) \geq y + \frac{1}{m}\sum_{i=1}^{m}\mathbb{E}H_i(x_i, x_i')\Big)$$

$$\leq c_1 \frac{V_p}{y^p} + \exp\Big(-c_2 \frac{y^2}{V_2}\Big), \tag{86}$$

*where for $q \in \{2, p\}$,*

$$V_q = \Big(1 + \frac{m}{N}\Big)^{q-1}\Big(1 + \log\Big(\frac{N}{u}\Big)\Big)$$

$$\times \sum_{i=1}^{m}\mathbb{E}\Big|2H_i(x_i, x_i') + \Big|\frac{1}{m} - \frac{1}{u}\Big|\beta' G(x_i, x_i')\Big|^q, \tag{87}$$

*and the constants $c_1 = 4(4\frac{p+2}{p})^p$ and $c_2 = 1/(2(p+2)^2 e^p)$ depend only on $p$.*

**Proof of Theorem C.4.** We do the similar decomposition as in Proof of Theorem C.4

$$G_i(x_i) := \mathbb{E}[\varphi \mid \mathcal{F}_{i-1}, x_i], \tag{88}$$

and

$$D_i = G_i(x_i) - \mathbb{E}[G_i(x_i) \mid \mathcal{F}_{i-1}]. \tag{89}$$

Let $R_{i-1} := X \setminus \{x_1, \ldots, x_{i-1}\}$ denote the remaining pool after the first $i-1$ draws, so $|R_{i-1}| = N - i + 1$. Conditional on $\mathcal{F}_{i-1}$, $x_i$ is uniform on $R_{i-1}$. Denote $x_{i=k}^{\ell} = \{x_k, x_{k+1}, \ldots, x_\ell\}$ and $S_1^m$ a sequence of random variables $S_1, \ldots, s_m$. We write $S_i^m = x_i^m$ as a shorthand for the m equalities $S_i = x_i, i = 1, \ldots, m$ and $\mathbb{P}(x_{i+1}^m | x_1^{i-1}, x_i) = \mathbb{P}(S_{i+1}^m = x_{i+1}^m | S_1^{i-1} = x_1^{i-1}, S_i = x_i)$. Let $x_i'$ be an independent copy of $x_i$ conditional on $\mathcal{F}_{i-1}$, which means conditional on $\mathcal{F}_{i-1}$, $x_i'$ is also uniform on $R_{i-1}$ and independent of $x_i$. Then $\mathbb{E}[G_i(x_i) \mid \mathcal{F}_{i-1}] = \mathbb{E}[G_i(x_i') \mid \mathcal{F}_{i-1}]$. Conditioning further on $(\mathcal{F}_{i-1}, x_i)$ gives the key identity

$$D_i = \mathbb{E}\left[G_i(x_i) - G_i(x_i') \mid \mathcal{F}_{i-1}, x_i\right]. \tag{90}$$

Then, consider fix $\mathcal{F}_{i-1}$ and fix $x_i, x_i' \in R_{i-1}$. Under $\mathcal{F}_{i-1}$ and $S_i = x_i$, the remaining coordinates $(S_{i+1}, \ldots, S_m)$ are sampled uniformly without replacement from $R_{i-1} \setminus \{x_i\}$; similarly under $S_i = x_i'$ they are sampled from $R_{i-1} \setminus \{x_i'\}$. By definition of conditional expectation, we have

$$D_i = \sum_{x_{i+1}^m} \phi(x_1^{i-1}, x_i, x_{i+1}^m)\mathbb{P}(x_{i+1}^m | x_1^{i-1}, x_i) - \sum_{x_{i+1}'^m} \phi(x_1^{i-1}, x_i', x_{i+1}'^m)\mathbb{P}(x_{i+1}'^m | x_1^{i-1}, x_i'). \tag{91}$$

Note that

$$\mathbb{P}(\mathbf{x}_{i+1}^m | \mathbf{x}_1^{i-1}, x_i) = \prod_{k=i}^{m-1} \frac{1}{N-i} = \frac{u!}{(N-i)!} = \mathbb{P}(\mathbf{x}'_{i+1}^m | \mathbf{x}_1^{i-1}, x_i'), \tag{92}$$

we can obtain

$$D_i = \frac{u!}{(N-i)!} \left( \sum_{\mathbf{x}_{i+1}^m} \phi(\mathbf{x}_1^{i-1}, x_i, \mathbf{x}_{i+1}^m) - \sum_{\mathbf{x}'_{i+1}^m} \phi(\mathbf{x}_1^{i-1}, x_i', \mathbf{x}_{i+1}^m) \right). \tag{93}$$

Consider the sequence of $\mathbf{x}_{i+1}^m$ and $\mathbf{x}'_{i+1}^m$, We can find bijection relationship among the two sequence: i) $\mathbf{x}_j = x_i', \mathbf{x}'_j = x_i$ and $\mathbf{x}_{i+1}^m \setminus \mathbf{x}_j = \mathbf{x}'_{i+1}^m \setminus \mathbf{x}'_j$; ii) $x_i' \notin \mathbf{x}_{i+1}^m, x_i \notin \mathbf{x}'_{i+1}^m$ and $\mathbf{x}_{i+1}^m = \mathbf{x}'_{i+1}^m$.

For the first case, for any fixed $j \in [i+1, m]$, the number of the permutations are $(m-i-1)! \binom{N-i-1}{m-i-1}$, thus we have

$$\left| \sum_{j=i+1}^m \left( \sum_{\mathbf{x}_{i+1}^m \setminus \mathbf{x}_j = \mathbf{x}'_{i+1}^m \setminus \mathbf{x}'_j} \phi(\mathbf{x}_1^{i-1}, x_i, \mathbf{x}_{i+1}^{j-1}, x_i', \mathbf{x}_{j+1}^m) - \sum_{\mathbf{x}_{i+1}^m \setminus \mathbf{x}_j = \mathbf{x}'_{i+1}^m \setminus \mathbf{x}'_j} \phi(\mathbf{x}_1^{i-1}, x_i', \mathbf{x}'_{i+1}^{j-1}, x_i, \mathbf{x}'_{j+1}^m) \right) \right|$$

$$\leq \sum_{j=i+1}^m (m-i-1)! \binom{N-i-1}{m-i-1} \left| \phi(\mathbf{x}_1^{i-1}, x_i, \mathbf{x}_{i+1}^{j-1}, x_i', \mathbf{x}_{j+1}^m) - \phi(\mathbf{x}_1^{i-1}, x_i', \mathbf{x}'_{i+1}^{j-1}, x_i, \mathbf{x}'_{j+1}^m) \right|$$

$$\leq \sum_{j=i+1}^m (m-i-1)! \binom{N-i-1}{m-i-1} \left( H_i(x_i, x_i') + H_j(x_i, x_i') \right). \tag{94}$$

For the second case, the number of those permutations are $(m-i)! \binom{N-i-1}{m-i}$, thus we can claim that,

$$\sum_{\mathbf{x}_{i+1}^m : x_i' \notin \mathbf{x}_{i+1}^m} \phi(\mathbf{x}_1^{i-1}, x_i, \mathbf{x}_{i+1}^m) - \sum_{\mathbf{x}'_{i+1}^m : x_i \notin \mathbf{x}'_{i+1}^m} \phi(\mathbf{x}_1^{i-1}, x_i', \mathbf{x}_{i+1}^m)$$

$$= (m-i)! \binom{N-i-1}{m-i} |\phi(\mathbf{x}_1^{i-1}, x_i, \mathbf{x}_{i+1}^m) - \phi(\mathbf{x}_1^{i-1}, x_i', \mathbf{x}_{i+1}^m)|$$

$$\leq (m-i)! \binom{N-i-1}{m-i} H_i(x_i, x_i'). \tag{95}$$

Combining above result, we can derive

$$|D_i| \leq \frac{u!}{(N-i)!} \left[ (m-i)! \binom{N-i-1}{m-i} H_i(x_i, x_i') + \sum_{j=i+1}^m (m-i-1)! \binom{N-i-1}{m-i-1} \left( H_i(x_i, x_i') + H_j(x_i, x_i') \right) \right]$$

$$\leq \frac{u}{N-i} H_i(x_i, x_i') + \sum_{j=i+1}^m \frac{1}{N-i} \left( H_i(x_i, x_i') + H_j(x_i, x_i') \right)$$

$$= H_i(x_i, x_i') + \sum_{j=i+1}^m \frac{1}{N-i} H_j(x_i, x_i') \tag{96}$$

Then, by similar proof as Theorem 2.2, we have

$$\mathbb{P}\left( \sum_{i=1}^m D_i > z \right) \leq 3 \left( 1 + \frac{2}{p} \right)^p \frac{S_p}{z^p} + \exp\left( - \frac{2}{(p+2)^2 e^p} \frac{z^2}{S_2} \right) \tag{97}$$

where

$$S_p = \sum_{i=1}^m \mathbb{E} \left| H_i(x_i, x_i') + \sum_{j=i+1}^m \frac{1}{N-i} H_j(x_i, x_i') \right|^p. \tag{98}$$

By Hölder's inequality, we can obtain

$$\mathbb{E}\left|H_i(x_i, x_i') + \sum_{j=i+1}^{m} \frac{1}{N-i} H_j(x_i, x_i')\right|^p \leq \left(1 + \sum_{j=i+1}^{m} \frac{1}{N-i}\right)^{p-1} \left[\mathbb{E}|H_i(x_i, x_i')|^p + \left(\frac{1}{N-i}\right) \sum_{j=i+1}^{m} \mathbb{E}|H_j(x_i, x_i')|^p\right]$$

$$\leq \left(1 + \frac{m}{N}\right)^{p-1} \left[\mathbb{E}|H_i(x_i, x_i')|^p + \left(\frac{1}{N-i}\right) \sum_{j=i+1}^{m} \mathbb{E}|H_j(x_i, x_i')|^p\right]. \qquad (99)$$

By taking sum over 1 to $m$,

$$S_p \leq \sum_{i=1}^{m} \left(1 + \frac{m}{N}\right)^{p-1} \left[\mathbb{E}|H_i(x_i, x_i')|^p + \left(\frac{1}{N-i}\right) \sum_{j=i+1}^{m} \mathbb{E}|H_j(x_i, x_i')|^p\right]$$

$$\leq \left(1 + \frac{m}{N}\right)^{p-1} \sum_{i=1}^{m} \left[\mathbb{E}|H_i(x_i, x_i')|^p + \sum_{j=1}^{m} \mathbb{E}|H_j(x_j, x_j')|^p \left(\sum_{i=j-1}^{m} \frac{1}{N-i}\right)\right]$$

$$\leq \left(1 + \frac{m}{N}\right)^{p-1} \left[\sum_{i=1}^{m} \mathbb{E}|H_i(x_i, x_i')|^p + \sum_{j=1}^{m} \mathbb{E}|H_j(x_j, x_j')|^p \times \log\left(\frac{N}{u}\right)\right]$$

$$\leq \left(1 + \frac{m}{N}\right)^{p-1} \left(1 + \log\left(\frac{N}{u}\right)\right) \sum_{i=1}^{m} \mathbb{E}|H_i(x_i, x_i')|^p \qquad (100)$$

The proof is completed.

$\square$

**Proof of Theorem C.5.** The result follows directly from Theorem C.4 and similar versions of Lemmas C.1 and C.2. $\square$

## C.3. Proof of Results in Section 3.3: Meta-Learning

**Proof of Theorem 3.14.** In the following, we suppress $\beta$, $\beta'$ and $\beta''$ inside the functions $H, G$ and $\mathcal{M}$. We follow a similar, but slightly more convoluted strategy compared to that of Theorem 3.4. Observe that, for $k \in [m], l \in [n]$,it holds almost surely that

$$mn\big|R(A(\mathbb{S}), \mathbb{S}) - R(A(\mathbb{S}^{(k,l)}), \mathbb{S}^{(k,l)})\big|$$

$$\leq \sum_{j\neq k}^{m} \sum_{i=1}^{n} |\ell(A(\mathbb{S})(\mathcal{S}_j), z_j^i) - \ell(A(\mathbb{S}^{(k,l)})(\mathcal{S}_j), z_j^i)| + \sum_{i\neq l}^{n} |\ell(A(\mathbb{S})(\mathcal{S}_k), z_k^i) - \ell(A(\mathbb{S}^{(k,l)})(\mathcal{S}_k^{(l)}), z_k^i)|$$

$$+ |\ell(A(\mathbb{S})(\mathcal{S}_k), z_k^l) - \ell(A(\mathbb{S}^{(k,l)})(\mathcal{S}_k^{(l)}), z_k^{l'})|$$

$$\overset{(a)}{\leq} (m-1)n\, H(z_k^l, z_k^{l'}) + \sum_{i\neq l}^{n} \Big(|\ell(A(\mathbb{S})(\mathcal{S}_k), z_k^i) - \ell(A(\mathbb{S})(\mathcal{S}_k^{(l)}), z_k^i)| + |\ell(A(\mathbb{S})(\mathcal{S}_k^{(l)}), z_k^i) - \ell(A(\mathbb{S}^{(k,l)})(\mathcal{S}_k^{(l)}), z_j^i)|\Big)$$

$$+ |\ell(A(\mathbb{S})(\mathcal{S}_k), z_k^l) - \ell(A(\mathbb{S}^{(k,l)})(\mathcal{S}_k), z_k^l)| + |\ell(A(\mathbb{S}^{(k,l)})(\mathcal{S}_k), z_k^l) - \ell(A(\mathbb{S}^{(k,l)})(\mathcal{S}_k^{(l)})), z_k^l)|$$

$$+ |\ell(A(\mathbb{S}^{(k,l)})(\mathcal{S}_k^{(l)}), z_k^l) - \ell(A(\mathbb{S}^{(k,l)})(\mathcal{S}_k^{(l)})), z_k^{l'})|$$

$$\overset{(b)}{\leq} mn\, H(z_k^l, z_k^{l'}) + n\, G(z_k^l, z_k^{l'}) + \mathcal{M}(z_k^l, z_k^{l'}), \qquad (101)$$

where $(a)$ follows from (9), and $(b)$ follows from (10) and (11) . On the other hand, it is trivial to note that

$$|R(A(\mathbb{S}), \mu) - R(A(\mathbb{S}^{(k,l)}), \mu)| \leq H(z_k^l, z_k^{l'}) \text{ almost surely.} \qquad (102)$$

Let $g \equiv R(A(\mathbb{S}), \mathbb{S}) - R(A(\mathbb{S}), \mu)$, and denote by $g_{k,l} \equiv R(A(\mathbb{S}^{(k,l)}), \mathbb{S}^{(k,l)}) - R(A(\mathbb{S}^{(k,l)}), \mu)$. Combining (101)-(102) provides that

$$|g - g_{k,l}| \leq 2\, H(z_k^l, z_k^{l'}) + \frac{1}{m}\, G(z_k^l, z_k^{l'}) + \frac{1}{mn} \mathcal{M}(z_k^l, z_k^{l'}). \qquad (103)$$

To deliver the coup de grâce via an application of Theorem 2.2, we are also required to control $\mathbb{E}_{\mathbb{S}}[g]$. To that end, we proceed as follows. Let $\mathbb{S}' = \{\mathcal{S}'_1, \ldots, \mathcal{S}'_m\}$ be an i.i.d. copy of $\mathbb{S}$, and $\mathbb{S}^{(j)} := \{\mathcal{S}_1, \ldots, \mathcal{S}_{j-1}, \mathcal{S}'_j, \mathcal{S}_{j+1}, \ldots, \mathcal{S}_m\}$. For each $i, j$, let $\mathbb{S}'' := \{z_j^{i''}\}$ be an i.i.d. copy of $\{z_j^i\}$, independent of $\mathbb{S}$ and $\mathbb{S}'$. Let $\mathcal{S}_j^{(i)} = \{z_j^1, \ldots, z_j^{i-1}, z_j^{i''}, z_j^{i+1}, \ldots, z_j^n\}$.

$$\mathbb{E}_{\mathbb{S},\mathcal{S},z}\Big[\frac{1}{mn}\sum_{j=1}^{m}\sum_{i=1}^{n}\big(\ell(A(\mathbb{S})(\mathcal{S}_j), z_j^i) - \ell(A(\mathbb{S})(\mathcal{S}), z))\big)\Big]$$

$$= \mathbb{E}_{\mathbb{S},\mathbb{S}',\mathbb{S}''}\Big[\frac{1}{mn}\sum_{j=1}^{m}\sum_{i=1}^{n}\big(\ell(A(\mathbb{S})(\mathcal{S}_j), z_j^i) - \ell(A(\mathbb{S}^{(j)})(\mathcal{S}_j^{(i)}), z_j^i))\big)\Big]$$

$$= \mathbb{E}_{\mathbb{S},\mathbb{S}',\mathbb{S}''}\Big[\frac{1}{mn}\sum_{j=1}^{m}\sum_{i=1}^{n}\big(\ell(A(\mathbb{S})(\mathcal{S}_j), z_j^i) - \ell(A(\mathbb{S}^{(j)})(\mathcal{S}_j), z_j^i) + \ell(A(\mathbb{S}^{(j)})(\mathcal{S}_j), z_j^i) - \ell(A(\mathbb{S}^{(j)})(\mathcal{S}_j^{(i)}), z_j^i))\big)\Big]$$

$$\leq \mathbb{E}_{\mathbb{S},\mathbb{S}',\mathbb{S}''}\Big[\frac{1}{mn}\sum_{j=1}^{m}\sum_{i=1}^{n}H(z_j^i, z_j^{i'}) + \frac{1}{n}\sum_{i=1}^{n}G(z_j^i, z_j^{i''})\Big]$$

$$= \mathbb{E}_{z,z'\sim\mathcal{D},\mathcal{D}\sim\mu}[H(z,z') + G(z,z')]. \tag{104}$$

Therefore, from Theorem 2.2, it follows that

$$\mathbb{P}\Big(R(A(\mathbb{S}),\mathbb{S}) - R(A(\mathbb{S}),\mu) \geq y + \mathbb{E}_{z,z'\overset{i.i.d.}{\sim}\mathcal{D},\mathcal{D}\sim\mu}[H(z,z') + G(z,z')]\Big) \leq c_1\frac{L_{p1}}{y^p} + 2\exp\big(-c_2\frac{y^2}{L_{p2}}\big), \tag{105}$$

where

$$L_{p1} := \sum_{k=1}^{m}\sum_{l=1}^{n}\mathbb{E}\big[\big(2\,H(z_k^l, z_k^{l'}) + \frac{1}{m}\,G(z_k^l, z_k^{l'}) + \frac{1}{mn}\mathcal{M}(z_k^l, z_k^{l'})\big)^p\big], \text{ and,}$$

$$L_{p2} := \sum_{k=1}^{m}\sum_{l=1}^{n}\mathbb{E}\big[\big(2\,H(z_k^l, z_k^{l'}) + \frac{1}{m}\,G(z_k^l, z_k^{l'}) + \frac{1}{mn}\mathcal{M}(z_k^l, z_k^{l'})\big)^2\big].$$

Evidently, using Hölder's inequality, it follows that

$$L_{p1} \leq C_p(mn\mathbb{E}[H^p] + \frac{n}{m^{p-1}}\mathbb{E}[G^p] + \frac{1}{(mn)^{p-1}}\mathbb{E}[\mathcal{M}^p])$$

$$L_{p2} \leq C_2(mn\mathbb{E}[H^2] + \frac{n}{m}\mathbb{E}[G^2] + \frac{1}{mn}\mathbb{E}[\mathcal{M}^2]),$$

which completes the proof in light of (105). $\qquad\square$

## D. Numerical Experiments

In this section, we provide empirical studies highlighting the tightness of our theoretical bounds in Sections 3.1 and 3.2. It is imperative we describe the general recipe of our experiments before going into the details. Since the applications in Section 3 stem from the key theoretical result Theorem 2.2, let us presume that the bounds in (3) are tight up to some constant. Therefore, when $z$ is large, the polynomial term $z^{-p}$ dominates in the decay, and from (3), it follows that

$$\frac{\mathbb{P}(|f(x_1, x_2, \ldots, x_n)| - \mathbb{E}[f(x_1, x_2, \ldots, x_n)]| > z)}{\mathbb{P}(|f(x_1, x_2, \ldots, x_n)| - \mathbb{E}[f(x_1, x_2, \ldots, x_n)]| > C_0 z)} \approx C_0^p. \tag{106}$$

On the other hand, in the absence of the polynomial term $z^{-p}$ or when $z$ is small, the sub-gaussian term dominates the tail probability, and consequently we should expect

$$\frac{\mathbb{P}(|f(x_1, x_2, \ldots, x_n)| - \mathbb{E}[f(x_1, x_2, \ldots, x_n)]| > z)}{\mathbb{P}(|f(x_1, x_2, \ldots, x_n)| - \mathbb{E}[f(x_1, x_2, \ldots, x_n)]| > C_0 z)} \approx 2\exp(C_1 z^2), \tag{107}$$

where $C_1$ is some constant depending on $p$, $\mathbb{E}[H_i]$ and $C_0$. The twin observations (106) and (107) informs our subsequent discussion, whereby any deviations from the expected behaviors would indicate our bounds are not sharp. In particular, we analyze the setting corresponding to Sections 3.1 and 3.2 in the following Sections D.1 and D.2, respectively.

## D.1. Simulations for empirical risk minimization

Consider i.i.d. observations $S := \{z_i = (x_i, y_i)\}_{i=1}^m \in \mathbb{R}^d \times \mathbb{R}$ from a linear model $Y = X\beta + \varepsilon$, where the errors $\varepsilon \overset{d}{=} U_1^{-1/\nu} - U_2^{-1/\nu}, U_1, U_2 \overset{i.i.d.}{\sim} U[0,1]$, is drawn from the distribution of the difference of two independent Pareto (type I) random variables. Note that here $p = \nu/2$. For our analysis, we take $d = 5$, $\beta = (1, 1, \ldots, 1)^\top$, and vary $m \in \{500, 1000\}$, and $\nu = \{2.2, 4.4\}$. For the scaling parameter $C_0$ in (106), we choose $C_0 = 1.5$.

To incorporate stability in our analysis, we resort to a ridge-regression with regularization parameter $\lambda = 1.0$; formally, let

$$\hat{\beta} = (X^\top X + \lambda I_d)^{-1} X^\top Y, \ X = (x_1 : \ldots : x_m)^\top, Y = (y_1, \ldots, y_m).$$

Note that $\hat{\beta}$ plays the role of $A_S$ in Section 3.1. Here we consider $(x_i)_{i=1}^m \overset{i.i.d.}{\sim} N(0, \Sigma), \Sigma_{ij} = 0.3^{|i-j|}$. We consider $\ell$ to be the squared error loss, so that

$$R(A, S) = \mathbb{E}_{x,y}[(y - x^\top \hat{\beta})^2], \quad R_{emp}(A, S) = \frac{1}{m} \sum_{i=1}^m (y_i - x_i^\top \hat{\beta})^2.$$

Elementary calculations show that Assumption 3.1 is satisfied with probability approaching 1 for $H(z_i, z_i') = |x_i y_i - x_i' y_i'|$ and $\beta = \frac{\log m}{m}$. Since $\beta$ is extremely small for large values of $m$, we may be excused for comparing $p(y) := \frac{\mathbb{P}(|R - R_{emp}| > y)}{\mathbb{P}(|R - R_{emp}| > C_0 y)}$ against $C_0^{\nu/2}$. The tail probabilities are empirically estimated via $50,000$ Monte Carlo draws. In Figure 3,

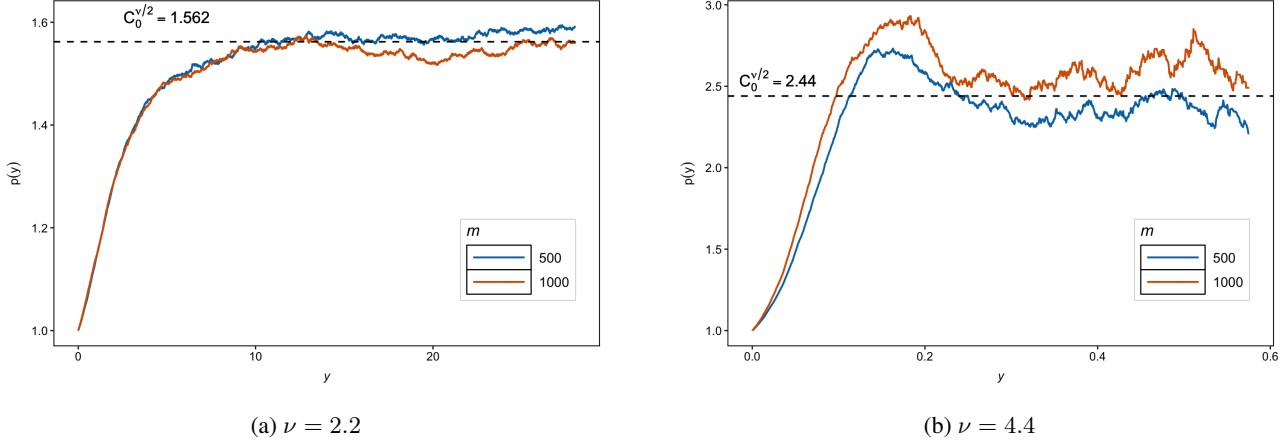

(a) $\nu = 2.2$          (b) $\nu = 4.4$

*Figure 3.* Plot of $p(y)$ versus $y$; both curves stabilize around $C_0^{\nu/2}$ for large $y$.

the ratio $p(y)$ exhibits an initial exponential growth before slowing down and stabilizing near $C_0^{\nu/2}$, further vindicating the behavior typified in (106) in light of Theorem 3.4. For larger $\nu$ (e.g., $\nu = 4.4$), the Gaussian tail dominates the polynomial tail more strongly at smaller values of $y$. Consequently, $p(y)$ may exceed the threshold $C_0^{\nu/2}$ initially, before stabilizing in the large-$y$ regime. This exhibits the tightness of our results.

For $\nu = 2.2$, we also fix $y = 30$, and plot $p(y)$ for $m \in \{500, 600, \ldots, 2000\}$. From Figure 4, it is evident that $p(y)$ remains quite stable around $C_0^{\nu/2}$ across a wider range of $m$ at fixed $y$. This confirms that the ratio is independent of $m$ for large $m$, justifying our original focus on varying $y$ to identify the sub-Gaussian and polynomial tails.

## D.2. Simulations for transductive regression algorithms

For the numerical studies involving transductive regression, we maintain the experimental set-up of ridge regression from Section D.1. Borrowing the notations of Section 3.2, let $S$ and $T$ denote the training and test set respectively, with $|S| = m$, and $|T| = u$. For the purpose of this experiment, we take $m = u$, and vary $m \in \{500, 1000\}$. Similar to Theorem C.5, we are concerned with

$$\hat{R}(h) = \frac{1}{m} \sum_{i \in S} (y_i - x_i^\top \hat{\beta})^2, \qquad R(h) = \frac{1}{m} \sum_{i \in T} (y_i - x_i^\top \hat{\beta})^2.$$

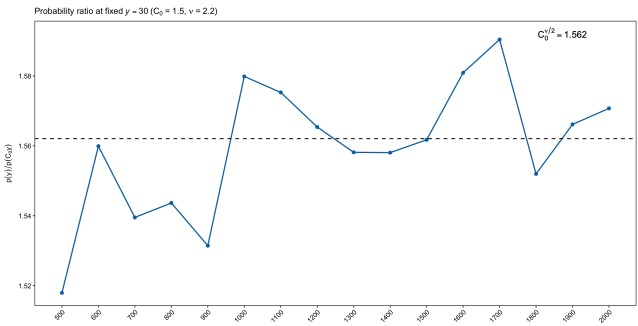

*Figure 4.* Plot of $p(y)$ versus $m$.

Note that here $\hat{\beta}$ is trained via ridge regression with $\lambda = 1.0$ *only* on the training sample $S$. Similar to Section D.1, tail probabilities are empirically estimated via $50{,}000$ Monte Carlo draws. Figure 5 showcases the corresponding line plots of $p(y)$ versus $y$, where $p(y) := \frac{\mathbb{P}(|R-\hat{R}|>y)}{\mathbb{P}(|R-\hat{R}|>C_0 y)}$. The stabilization of the curve $p(y)$ around the threshold $C_0^{\nu/2}$ is evident for this example as well, exhibiting the sharpness of our bounds.

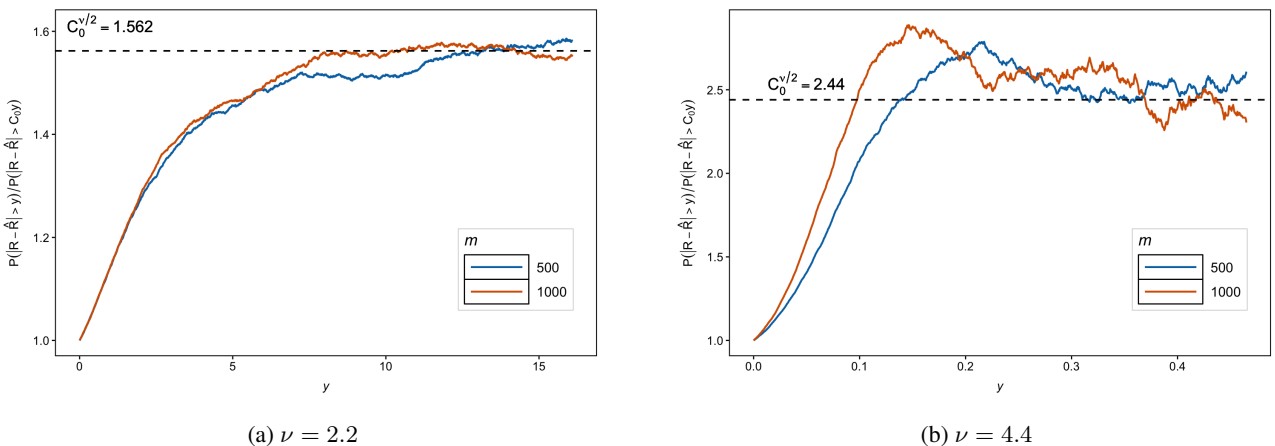

(a) $\nu = 2.2$             (b) $\nu = 4.4$

*Figure 5.* Plot of $p(y)$ versus $y$ for the transductive regression problem in Section 3.2.

### D.3. Additional experiments: a modern application.

In principle, any data with heavy tail would exhibit behavior similar to what we predict in our theory. Nevertheless, as baby steps towards more grounded stability theory, we perform a similar experiment on two layer neural-network for a regression problem. Specifically, we consider the set-up of Section D.1 in the current manuscript; consider i.i.d. observations $S := \{z_i = (x_i, y_i)\}_{i=1}^m \in \mathbb{R}^d \times \mathbb{R}$ from a linear model $Y = X\beta + \varepsilon$, where the errors $\varepsilon \overset{d}{=} U_1^{-1/\nu} - U_2^{-1/\nu}$, $U_1, U_2 \overset{i.i.d.}{\sim} U[0,1]$, is drawn from the distribution of the difference of two independent Pareto (type I) random variables. Note that here $p = \nu/2$. For our analysis, we take $d = 5$, $\beta = (1, 1, \ldots, 1)^\top$, and vary $m \in \{500, 1000\}$, and $\nu = \{2.2, 4.4\}$. For the scaling parameter $C_0$, we choose $C_0 = 1.5$.

To $S$, we fit a two-layer neural net

$$\mathcal{A}_S : x \mapsto W_2 \, \mathrm{ReLU}(W_1 x + b_1) + b_2, \ W_1 \in \mathbb{R}^{32 \times d}, W_2^\top \in \mathbb{R}^{32}, b_1 \in \mathbb{R}^{32}, b_2 \in \mathbb{R}.$$

The neural net is fitted with an Adam optimizer with learning rate $10^{-3}$ and 200 epochs. Thereafter, we evaluate the empirical loss and the true loss as in Section 3.1. The tail probabilities are estimated via $10{,}000$ Monte Carlo repetitions. Finally, similar to Section D.1, we plot $p(y)$ versus $y$, where the desired threshold is $C_0^{\nu/2} \approx 1.562$.

We see that even for a training algorithm such as a Neural net, optimized via Adam, the heavy-tailed nature of the data shines

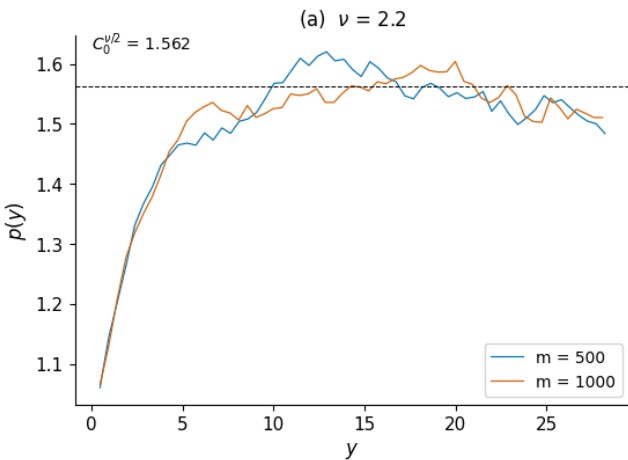

*Figure 6.* Plot of $p(y)$ versus $y$ for the neural network regression experiment in Section D.3.

through in distinct characterizations of Gaussian and polynomial tails. As predicted by our theory and can also be seen for the simpler ridge-regression example, $p(y)$ stabilizes at around $C_0^{\nu/2} \approx 1.562$. This corroborates our theory on a prototype of modern ML algorithm. Note that here, the number of parameters being optimized using Adam is also comparable to the sample sizes $m$, which further indicates the applicability of our results in modern, large-dimensional scenarios.

