# OpenReview forum: "Stability beyond Bounded Differences: Sharp Generalization Bounds under Finite $L_p$ Moments"
_ICML.cc/2026/Conference — ICML 2026 regular_

### Official Review · Reviewer_rYTF · 2026-02-26

**Soundness:** 3
**Presentation:** 3
**Significance:** 2
**Originality:** 3
**Overall Recommendation:** 4
**Confidence:** 4

**Summary:**

The main result of this paper is a new Mcdiarmid-type concentration inequality, under the boundedness of the moments of difference function, which is an assumption weaker than the classical bounded difference property, as well as the boundedness of moment generating function studied previously in this line of researches. The newly-derived concentration inequality captures a fine-grained behavior of the tail probability under such weaker assumptions, that is, a sub-Gaussian tail for small deviations and a polynomial decay for large deviations. The paper also discusses its applications to several learning problems for their new probabilistic tool, showing generalization guarantees under weaker assumptions. Finally, the authors provide empirical justifications to their theory.

**Compliance With Llm Reviewing Policy:**

Affirmed.

**Final Justification:**

I would recommend a score 4 with confidence 4 to this paper. The following are my evaluation on this work.

(i) Soundness: All the theorems stated in this paper are supported with rigorous proofs. The authors also did numerical experiments to justify their theoretical results, making the entire structure look complete. However, I do agree with the other reviewers' comments that, it would be good to include more experiments for those applications discussed in Section 4.

(ii) Presentation: Overall, I think the paper is well-written with a clear logic of presenting the newly-derived concentration inequality first, discussing potential applications in learning theory next, and complementing the theory with empirical demonstrations finally. I especially enjoy their theorems' statements, where the bounds are clean and the assumptions are clear. The paper also includes components of main contributions, related works as well as conclusion and future works.

(iii) Significance: This might be one of the parts that I will have concerns on. Taking both the original submission and the authors' rebuttal into consideration, I still believe the results could be more interesting and find more applications in machine learning scenarios if the new concentration inequality derived can be used to improve generalization guarantees for certain applications. Indeed, their fine-grained tail bounds could potentially lead to an improvement on the sample complexity for certain learning problems or even a better understanding of their non-asymptotic convergence rates' behaviors. But I would like to mention that the authors' rebuttal convince me  in some sense on the significance of relaxing the assumptions of sub-Gaussian and sub-exponential distributions.

(iv) Originality: After reading the authors' rebuttal, I decide to increase the score of originality as the results do seem to provide new insights into the area of stability-based generalizations. I also realize some parts of the technical novelty of their proofs.

Putting together, I think this work is interesting but hope the authors could make additional experiments and further polish the paper.

**Key Questions For Authors:**

Can the authors explain the following, which are important to my evaluation of the paper. I would be happy to raise my score if convinced.
1. I highly respect the technical efforts presented in this work but still have a confusion. What is the technical novelty of proving this concentration inequality compared to the proof of those variants in previous works, especially the one in https://proceedings.mlr.press/v235/li24cs.html. In other words, is a simple modification on some parts of the original proof suffices or a novel proof technique is required? If latter, what makes the original proof strategy fail? Intuitively (from my point of view), when the boundedness of MGF replaced by the boundedness of moments, one can try to adapt a weaker version of the concentration inequality into the analysis. A even more naive idea would be to show first a high probability uniform boundedness and then plug into the classical McDiarmid-type inequality together with a union bound. Hence, I don't fully understand the difficulty of the analysis.

2. A probabilistic tool is usually more interesting and impactful if it helps to establish tighter guarantees for a statistical problem under the same assumptions. In this paper, the benefit of the new concentration inequality is, in contrast, to help relax the assumptions. Let us take Section 3.1 as an example. The work of https://arxiv.org/abs/1910.07833 gives a tight generalization guarantee under uniform stability and the uniform boundedness of loss function, and Theorem 3.3 provide generalization guarantee under weaker assumptions. What kind of practical learning scenarios using ERM can potentially benefit from this relaxation? I feel that sub-gaussian and sub-exponential assumptions are indeed practical (please correct me if you don't agree). Another interesting direction would be to provide lower bounds under those weak assumptions and evaluate the optimality of the concentration analysis.

**Limitations:**

Yes.

**Strengths And Weaknesses:**

Strengths:
The newly-derived concentration inequality is interesting and requires careful technical analysis. According to my understanding, the main result of this paper seems likely to be a fine-grained concentration probability control, that is, for a large deviation the failure probability decreases in a polynomial rate, which is indeed a fair contribution.

Weakness:
Besides the questions stated in the following section, I noticed several minor issues:
1. Equation (2) is a bit confusing: is it $e^{(|X|/\theta)^{\alpha}}$ or $(e^{|X|/\theta})^{\alpha}$?
2. In Assumption 3.1, both $H:\mathcal{Z}\times\mathcal{Z}\mapsto\mathbb{R}$ and $\lVert H(z)\rVert_{p}<\infty$ were written, I'm confused about the domain of the function $H$.
3. In line 271, the authors stated that "Assumption 3.1 significantly generalizes the Lipschitz stability notion of ...", but I didn't find a discussion about this claim. It would be even nicer if the authors can identify certain practical learning scenarios where such assumptions are commonly satisfied but the previous ones are not.
4. Following the above bullet 3, in line 256, the authors mentioned the example of regression with unbounded responses. The question is , can't a generalization guarantee be proved based on the concentration inequality in https://proceedings.mlr.press/v235/li24cs.html? Because it also handles the case of unbounded loss.

---

> ### Author Rebuttal · Authors · 2026-03-31
>
> We thank you for the detailed feedback and address your concerns sequentially below. We also thank you for considering to increase our score upon a successful rebuttal.
>
> **Ans to Q1+W4:** We agree that the technical distinction from Li et al. [1] should be clarified. The key point is that our result is not obtained by simply weakening Thm 2.2 in [1]. Their proof relies primarily on Markov inequality applied to moment bounds, a strategy that is technically limited and insufficient to establish the two-regime concentration result. Markov inequality yields at best a polynomial bound of order $E|f-E(f)|^p/t^p$. Even in the simplest case $X_i\sim N(0,1)$, $f(X_1,...,X_n)=\sum_{i=1}^nX_i$, this gives order $\frac{n^{p/2}}{t^p}$, which is worse than both terms in our Thm 2.2. Thus our result cannot be recovered by simple modifications to their proof.
>
> Our proof decomposes the martingale difference sum into a truncated part and a large-jump part. The large-jump part is controlled via Markov inequality using only the available p-th moments, while the truncated part is handled through boundedness of the MGF - a mandatory ingredient that cannot generally be replaced by moment bounds alone. This MGF step is fundamental for recovering the sharp sub-Gaussian component and is precisely the mechanism underlying classical Bernstein-type inequalities.
>
> We agree that the intuition of combining a union bound with McDiarmid inequality is related to our proof strategy; indeed, our MGF argument for the truncated part is very much in the spirit of standard McDiarmid/Azuma proofs. However, a direct application of McDiarmid after truncation is not optimal in our setting. The novelty lies in bounding the MGF directly rather than applying McDiarmid as a black box, which allows us to recover the optimal two-regime bound as explained above. Finally,  [6], Cor. 29 states a result similar to our Thm 2.2, but its proof contains a fatal gap, further suggesting the non-triviality of our arguments. We refer the details to our rebuttal to W1 of reviewer bPai.
>
> **Ans to Q2+W3:** We agree that stronger probabilistic tools are compelling when they improve guarantees under the same assumptions. Our contribution differs in emphasis: we show that meaningful high-probability generalization guarantees remain possible under substantially weaker conditions, motivated by the fact that sub-Gaussian and sub-exponential assumptions are not always adequate for learning problems with unbounded or heavy-tailed data.
>
> As a direct example, ridge regression is a regularized ERM problem already known to satisfy higher-moment stability (see [2]), as we also demonstrate numerically. The same literature identifies k-nearest neighbors as a canonical stable algorithm, showing such assumptions are natural beyond ERM. More broadly, the relaxation is relevant for modern learning with heavy-tailed or unbounded data. For instance, [3] on ERM/DP-SGD studies multinomial logistic regression and last-layer training atop pretrained networks under sample-dependent Lipschitz envelopes with only bounded moments, and explicitly notes that uniform Lipschitzness may fail even for simple linear regression with Gaussian covariates. Our new neural-network experiments trained with ADAM provide another modern practical example, and their results further support our theoretical findings (Refer to our rebuttal to Reviwer Sk3C Q5). In such settings, the uniformly bounded loss and uniform stability assumptions of [4] are often unavailable, motivating weaker moment-based conditions.
>
> The two-regime form is not merely an upper-bound artifact. Related work [5] establishes an exact asymptotic equality for linear processes, not just an upper bound. Moreover, with suitable adaptation of the techniques in [5] and its references, a corresponding equality result can be derived in our setting, further confirming the optimality of our bound at the level of tail form and order. At your insistence, we are ready to add a remark to clarify this point in the camera-ready version.
>
> **Ans to W1:** The intended expression in Eqn (2) is $E(e^{(|X|/\theta)^\alpha})\leq 2$, i.e., the standard definition of the $\psi_\alpha$-Orlicz norm. We have corrected the parentheses in revision.
>
> **Ans to W2:** This is a typo in Assumption 3.1: the intended object is $H(z,z')$, defined on $\mathcal{Z}\times\mathcal{Z}$, and the $L_p$ requirement is $\\|H(z,z')\\|_p<\infty$. We will fix this in revision.
>
> **Reference**
>
> [1] Algorithmic Stability Unleashed: Generalization Bounds with Unbounded Losses
>
> [2] Stability revisited: new generalisation bounds for the Leave-one-Out.
>
> [3] Beyond Uniform Lipschitz Condition in Differentially Private Optimization
>
> [4] Sharper Bounds for Uniformly Stable Algorithm
>
> [5] Exact Moderate and Large Deviations for Linear Processes
>
> [6] Concentration and Moment Inequalities for General Functions of Independent Random Variables with Heavy Tails.

---

> > ### Author Rebuttal · Reviewer_rYTF · 2026-04-03
> >
> > I thank the authors for providing me with detailed responses to my questions and concerns. I especially like your responses to my questions Q1 and Q2. Most of my concerns have also been addressed appropriately in this rebuttal. Given the fact that the paper overall makes fair contributions towards understanding generalization as well as this successful rebuttal, I decide to raise my score.

---

> > > ### Author Response · Authors · 2026-04-05
> > >
> > > Thank you very much for your thoughtful follow-up and positive reassessment of our paper. We sincerely appreciate your careful reading and your encouraging comments, especially regarding our responses to Q1 and Q2. Your feedback has been very helpful in improving the paper.

---

### Official Review · Reviewer_Sk3C · 2026-03-01

**Soundness:** 2
**Presentation:** 2
**Significance:** 3
**Originality:** 3
**Overall Recommendation:** 3
**Confidence:** 4

**Summary:**

This paper studies high-probability generalization guarantees from the viewpoint of algorithmic stability under heavy-tailed or unbounded losses. The main idea is to replace classical bounded-differences / sub-Gaussian / sub-Weibull assumptions with an (L_p,β)-Lipschitz stability condition, requiring only finite p-th moments of one-sample perturbations. The technical core is a new concentration inequality for functions of independent variables under finite L_p increments, with a two-regime tail behavior: a sub-Gaussian term for moderate deviations and a polynomial term for large deviations. The paper then applies this framework to three learning settings—standard ERM, transductive regression, and meta-learning—and provides synthetic simulations intended to illustrate the sharpness of the predicted heavy-tail correction.

**Compliance With Llm Reviewing Policy:**

Affirmed.

**Key Questions For Authors:**

1. In Theorem 3.13 / Remark 3.14, can you clarify the exact additive shift and all coefficients (especially the roles of "𝛽,𝛽′,𝛽" in the expectation term and in the M-term)? If these are typographical issues, please provide the corrected statement. A precise correction here would materially increase my confidence in the paper’s soundness.

2. Which parts of the concentration arguments are fundamentally new, and which are adaptations of existing Nagaev-type / martingale-decomposition techniques? A sharper comparison to prior proof strategies would help me better assess originality.

3. Can you provide at least one concrete modern learning setting (beyond ridge regression in synthetic heavy-tailed linear models) where the proposed (L_p,β)-stability assumptions can be verified or reasonably argued to hold? A convincing example would improve my assessment of significance.

4. The paper repeatedly uses the term “sharp.” Do you mean sharp in exponent/order, up to constants, or in a stronger minimax sense? Please state precisely in which sense each main theorem is claimed to be sharp. This would affect how strongly I interpret the novelty and technical contribution.

5. Can you expand the experiments to include either real heavy-tailed data or a broader set of algorithms/settings? Stronger empirical evidence would improve my assessment of significance and help justify the paper’s framing for a broad ML audience.

**Limitations:**

yes

**Strengths And Weaknesses:**

Strengths.
1. The problem is important and timely. Stability-based generalization theory is classically strongest under bounded losses or exponential-tail assumptions, and extending it to finite-moment heavy-tailed regimes is a meaningful theoretical direction.
2. The paper has a coherent technical arc. Section 2 develops a general concentration tool, and Section 3 instantiates it in three nontrivial learning scenarios. This gives the work a unified narrative rather than a collection of disconnected lemmas.
3. The without-replacement extension for transductive regression is a nontrivial addition beyond the i.i.d. setting. This part is conceptually interesting and is likely the most distinctive application section.

Weaknesses.

1. Soundness / technical confidence is weakened by inconsistencies in the statement of the meta-learning result. In Theorem 3.13 and Remark 3.14, the additive shift and some coefficients appear inconsistent (e.g., the theorem statement uses y+E[H+G], while the high-probability form uses E[βH+β′G]; the treatment of the M-term is also not fully consistent in notation). These may be typographical issues, but they occur in a central theorem and make it harder to trust the details without a very careful line-by-line verification.

2. The “sharpness” claim is somewhat overstated. The paper does provide intuition and a canonical heavy-tailed example supporting the necessity of the polynomial term, but for the broader set of results the evidence is mostly asymptotic or illustrative rather than fully matched lower bounds. In that sense, the results look plausible and well-motivated, but “sharp” is stronger than what is fully established throughout the paper.

3. Empirical evaluation is limited. The experiments are synthetic and mainly verify that a tail-ratio curve stabilizes in a way consistent with the theory. This is useful as a sanity check, but it is not enough to demonstrate practical relevance or to show that the assumptions can be instantiated in realistic ML pipelines. There are no real datasets, no modern optimization examples, and no empirical validation of the proposed stability moments in practice.

4. The paper remains somewhat abstract at the level of applicability. The authors are honest that weaker tail assumptions come with stronger decay requirements on stability parameters. However, the paper does not convincingly show that these requirements are met by concrete modern learning algorithms beyond stylized ridge-regression-type examples.

5. Presentation needs tightening. The overall structure is clear, but the paper would benefit from a cleaner delineation of what is genuinely new in the proof techniques versus what is adapted from classical Nagaev-type or martingale arguments. In a theory paper, this distinction is important for assessing novelty.

Overall, I think the paper has clear merit and a respectable theoretical contribution. However, in its current form, the combination of central-statement inconsistencies and limited empirical/instantiation support keeps it below the acceptance bar for a top-tier broad ML venue.

---

> ### Author Rebuttal · Authors · 2026-03-31
>
> We thank the reviewer for the detailed feedback and address the concerns sequentially below. We also humbly request the reviewer to increase the score if all concerns have been addressed.
>
> **Ans to Q1+W1:** Indeed, $E[H+G]$ is a typo; it should be $E[\beta H+\beta'G]$. Similarly, in Rmk 3.14, the correct term is $\beta''\\|\mathcal{M}\\|_2$ instead of $\\|\mathcal{M}\\|_2$. We apologize for the error and have corrected both. We also proofread the manuscript carefully for further typos.
>
> **Ans to Q2+W5:** While inspired by Nagaev-type ideas, our setting involves neither independence nor a simple sum, so original Nagaev inequalities do not directly apply. A naive truncation + McDiarmid approach is suboptimal and also not applicable: truncation destroys the martingale structure, and uniform boundedness of the truncated part does not recover the sharp tail behavior. Instead, we decompose the martingale-difference sum into a truncated part and a large-jump part, control the latter by moment bounds, and handle the former via MGF arguments---the same mechanism underlying McDiarmid/Azuma-Hoeffding. This more delicate approach is essential; as noted in our reply to Reviewer bPai, prior attempts [1] at similar results contain a fatal gap, underscoring the need for the techniques developed here.
>
> **Ans to Q3+W4:** Beyond ridge regression, several concrete settings fit our framework. First, [2] gives a nearest-neighbor-type regression example in an unbounded metric space with stronger, sub-Gaussian-type stability control. Second, our assumptions subsume classical uniform stability: taking $H\equiv1$ recovers standard examples such as Hilbert-space regularization, SVMs, and related regularized ERM procedures from [3]. Third, the unbounded-loss, sub-Weibull setting of [4] also lies inside our setup, since sub-Weibull control implies finiteness of all moments, whereas we require only one fixed p-th moment. Finally, we give an evidence with Adam:  [the corresponding Figure](https://github.com/stabilityalgorithmicml-prog/Algorithm_stability_ICML2026/blob/main/new_Experiment_NeuralNet_ADAM_2.2.png) exhibits the same two-regime behavior predicted by our theory. Together with the broader stability literature for SGD beginning with [5], this suggests that our $(L_p,\beta)$ framework is a natural finite-moment extension of stability theory to heavy-tailed regimes.
>
> **Ans to Q4+W2:** We use "sharp" in the tail-order, not minimax sense. For Thm 2.2, under a finite $L_p$ condition, the correct deviation form is sub-Gaussian plus polynomial (Rmk 2.4). Refer to our reply to Q2 of reviewer rYTF, where we discuss that our tail bound achieves the precise order, not merely an upper bound. The same interpretation applies to Section 3. This two-regime behavior, motivated by the Fuk-Nagaev inequality, is optimal in the classical probability literature for sums of iid random variables, and our experiments suggest the same holds for stability analysis. Thus the novelty is the Gaussian-plus-polynomial deviation structure under weaker $L_p$ stability assumptions. This constitutes the key insight and the backbone of our technical contribution, since deriving such results is nontrivial: while responding to Reviewer bPai's comments, we identified that [1] attempts a similar result in Cor 29, but their proof contains a fundamental and irreparable gap. This underscores both the importance of the problem and the technical difficulty involved.
>
> **Ans to Q5+W3:** Our paper is mainly theoretical, and most stability papers provide no numerical experiments on the tightness of tail bounds (eg [4] [6] [7]), since the constants are unknown and problem-dependent. Our paper already gives a sanity check via the ratio of tail probabilities at $y$ and $C_0y$, which cancels these constants and isolates dependence on $C_0$. Still, broader experiments are valuable. We add a further experiment: under the same heavy-tailed linear model as Section C.1, we train a two-layer ReLU network, each with $32$ hidden layers, where the training is done via Adam. [The corresponding Figure](https://github.com/stabilityalgorithmicml-prog/Algorithm_stability_ICML2026/blob/main/new_Experiment_NeuralNet_ADAM_2.2.png) shows the ratio of large deviation probabilities stabilizing near the predicted threshold, with the same Gaussian-plus-polynomial behavior as in ridge example, indicating applicability to modern stochastic optimization algorithms. We have added this example in the revision.
>
> **Reference**
>
> [1]Concentration and Moment Inequalities for General Functions of Independent Random Variables with Heavy Tails.
>
> [2]Concentration in unbounded metric spaces and algorithmic stability.
>
> [3]Stability and generalization.
>
> [4]Algorithmic Stability Unleashed: Generalization Bounds with Unbounded Losses.
>
> [5]Train faster, generalize better: Stability of stochastic gradient descent.
>
> [6]Algorithmic stability and hypothesis complexity.
>
> [7]On the stability and generalization of meta-learning.

---

### Official Review · Reviewer_bPai · 2026-03-11

**Soundness:** 3
**Presentation:** 3
**Significance:** 2
**Originality:** 2
**Overall Recommendation:** 3
**Confidence:** 3

**Summary:**

This paper presents a stability-based framework that provides generalization bounds under finite $L_p$ moment conditions. It derives sharp concentration inequalities for functions of independent random variables with $L_p$ constraints. Furthermore, the paper applies these bounds to empirical risk minimization, transductive regression, and meta-learning, thereby expanding the scope of stability analysis in modern data science tasks, particularly those involving heavy-tailed data or unbounded losses.

**Compliance With Llm Reviewing Policy:**

Affirmed.

**Key Questions For Authors:**

1) The paper emphasizes that the proposed generalization bound is sharper. However, in Theorem 3.3, the term $\beta \|H\|_2 \sqrt{m}$ seems potentially difficult to control, where $m$ denotes the sample size. Further clarification on how this term can be effectively bounded would strengthen the theoretical claims.
2) Some writing issues should be addressed for clarity. In particular, the symbols $m$ and $n$ both appear to denote the total sample size. In Assumption 3.1, $n$ is used, whereas in Theorem 3.3, $m$ is used. If there is a substantive distinction between $m$ and $n$, this difference should be explicitly clarified. Otherwise, it would be preferable to unify the notation to improve readability and consistency.
3) The experimental design is insufficient to support claims of sharpness. Using only 50,000 Monte Carlo samples and two training sizes (m = 500, 1000) does not allow reliable estimation of tail probabilities or assessment of asymptotic scaling. Heavy-tailed deviations are high-variance events, and ratio-based metrics further amplify noise. With only two m values, the predicted dependence on sample size cannot be verified. More extensive scaling experiments and direct comparisons between empirical quantiles and theoretical bounds are necessary to substantiate tightness claims.

**Limitations:**

see weaknesses and questions

**Strengths And Weaknesses:**

Strength:
The paper proposes a comprehensive theoretical framework and provides a thorough discussion of high-probability generalization bounds. The analysis is detailed, covering multiple regimes of $p$ and clearly decomposing the polynomial and sub-Gaussian tail behaviors. The theory is systematically extended to several learning settings, demonstrating technical depth and internal consistency.

Weaknesses：
The manuscript appears to lack a detailed discussion of closely related literature. For example, Reference [1] introduces an $L_p$ stability framework, and Reference [2] discusses $L_p$ moment conditions in Theorem 4, Theorem 5, and Remark 6. The authors are encouraged to provide a more thorough comparison with these closely related works in order to clearly delineate the novelty of the present results.
[1] Stability revisited: new generalisation bounds for the Leave-one-Out
[2] Concentration and Moment Inequalities for General Functions of Independent Random Variables with Heavy Tails

---

> ### Author Rebuttal · Authors · 2026-03-31
>
> We sincerely appreciate your careful evaluation. We respectfully ask you to consider raising the score if the concerns have been satisfactorily addressed.
>
> **Ans to W1:** We connect the two papers to our work. We have added an extended discussion in our paper.
>
> [1] and our work differ in emphasis: [1] develops and verifies $L_q$-stability as a higher-moment stability notion for concrete models, whereas our paper establishes high-probability guarantees under an $L_q$-moment stability assumption. Thus, [1] provides complementary model-side evidence that our assumptions are meaningful, further motivating our work.
>
> [2] is closely related. Their Thm 4, 5 and Rmk 6 establish an $L_q$-moment bound used toward their Cor 29. While Cor 29 states a result similar to our Thm 2.2, its proof contains a gap: it first applies Markov inequality, which yields at best a polynomial bound of order $E|f-E(f)|^p/t^p$. Even in the simple case $X_i\sim N(0,1)$, $f(X_1,...,X_n)=\sum_{i=1}^n X_i$, this gives order $n^{p/2}/t^p$, far worse than claimed conclusion. Moreover, the argument yields a bound only at a particular threshold $t$, and cannot be extended to all $t>0$. Instead, our Thm 2.2 provides direct, self-contained derivation of the tail bound under the stated finite-moment assumption.
>
> **Ans to Q1:**
> Our Thm 2.2 and subsequent results are sharp under the weakened conditions due to the coexistence of polynomial and sub-Gaussian tails and supported by simulation (also see our rebuttal to Q2+Q3 of reviewer rYTF). In many applications, controlling $\beta\|H\|\_2\sqrt{m}$ is straightforward. As noted in Section 3.1, typically $\beta\ll1/\sqrt{m}$. The $\sqrt{m}$ factor is standard: [3, Thm 2.12] also contains a $\gamma\Delta_\alpha\sqrt{m\log(1/\delta)}$ term, and that paper notes stability coefficients are often of order $m^{-1/2}$. For ridge regression (Section C.1), $\beta=\log m/m$, and $H(z_i,z_i')=|x_iy_i-x_i'y_i'|$ with high probability, so $\\|H\\|_2=\sqrt{E[H^2]}=O(1)$. Similar intuition applies more broadly, including the neural net+Adam example in our rebuttal to Q5 of Reviewer Sk3C. In contrast, the uniform stability assumption in [4] requires $H\asymp1$, which typically needs bounded covariates and responses. Our framework allows unbounded covariates and responses while still yielding informative rates.
>
> **Ans to Q2:** There is no substantive distinction between $m$ and $n$ in Section 3.1: both denote the sample size in the i.i.d. ERM setting. This is only a notational inconsistency, and we will unify it in revision.
>
> **Ans to Q3:**
> - reliable estimation of tail probabilities: We respectfully disagree that 50,000 Monte Carlo samples are insufficient. For $m=500$ in the ridge regression example with $t_{2.2}$ errors, we repeated estimation of $P(|R-\hat{R}|>y)$ over 50,000 runs, where $y=30$ is approximately the 95\% quantile of $|R-\hat{R}|$. The empirical mean is 0.047 and the standard deviation is 0.00101. Under a Bin$(50,000,0.05)$ model, the standard deviation is $\approx0.000975$, closely matching the observed value and confirming that 50,000 repetitions are adequate.
> - Sample Size: The experiment is not meant to verify dependence on $m$, but to the $p$-dependent tail exponent. Since $p=\nu/2$, the ratio $P(|R-\hat{R}|>y)/P(|R-\hat{R}|>C_0y)$ should stabilize near $C_0^{\nu/2}$ in the large-deviation regime. Thus, the goal is to assess sharpness in $p$ and the necessity of the polynomial term; the two values of $m$ serve only as a robustness check. Nevertheless, per your suggestion, we added an experiment (see [Figure here](https://github.com/stabilityalgorithmicml-prog/Algorithm_stability_ICML2026/blob/main/new_experiment_ratio_vs_m.png) ) showing that above ratio remains stable around $C_0^{\nu/2}$ over a wider range of m at fixed $y$. This confirms that the ratio is independent of $m$ for large $m$, justifying our original focus on varying $y$ to identify the sub-Gaussian and polynomial tails.
> - Scaling: As a scaled-up experiment, we train a two-layer neural network via Adam on the same linear model setup. Refer to our reply to Q5 of Reviewer Sk3C for full details.
> - Direct Comparison: We respectfully note that our paper is primarily theoretical, and most stability literature include no numerical validation of tail bounds (eg [1],[2],[3]). Such validation is challenging because the bounds contain unknown, problem-dependent constants. To the best of our knowledge, our paper provides one of the **first numerical results** substantiating the theory. We achieve this by taking the ratio of large-deviation probabilities at $y$ and $C_0y$, which cancels all unknown constants and isolates a function of $C_0$ alone. A direct comparison of bound versus empirical probability would not be meaningful precisely because these constants are unavailable.
>
> [1] [2] are the papers you suggested.
> [3] Algorithmic Stability Unleashed: Generalization Bounds with Unbounded Losses.
> [4] Sharper bounds for uniformly stable algorithms.

---

> > ### Author Rebuttal · Reviewer_bPai · 2026-04-03
> >
> > Thanks for the detailed response.  I keep my initial score.

---

> > > ### Author Response · Authors · 2026-04-08
> > >
> > > Thank you for the acknowledgment. Since the remaining concern was not specified, we focus on the two points most relevant to originality and significance: the technical novelty of the theory and the new numerical evidence supporting its practical relevance.
> > >
> > > First, compared with [1], which provides model-side evidence that higher-moment stability can hold for concrete algorithms, our paper derives high-probability generalization guarantees under this weaker assumption. Moreover, the arguments in [2] are based only on moment bounds plus Markov inequality, which does not establish the full all-$t$ statement as written. *In contrast*, our proof uses a nontrivial truncation-plus-MGF decomposition: finite $L_p$ moments control the large-jump part, while a direct MGF bound for the truncated part recovers the sharp sub-Gaussian term. This is the key technical step behind the precise two-regime tail order, and it also explains why a superficially similar statement in [2] and [3] is not established by the proof as written.
> > >
> > > Importantly, the two-regime form that we obtain has the precise tail order: it captures the correct asymptotic behavior rather than being only a loose upper estimate. Note that the related work [4] establishes an exact asymptotic equality for analogous linear processes, and the same mechanism can be adapted to our setting. In this sense, our theorem should be understood as identifying the correct two-regime tail form and order, not merely proposing one possible upper bound.
> > >
> > > More broadly, going beyond a concentration theorem, our work builds a stability-based framework that turns this result into high-probability generalization bounds for i.i.d. ERM, transductive regression, and meta-learning under finite-moment stability. The relevant assumptions are substantially weaker than bounded-loss or uniform-stability conditions, and are tailored to learning problems with unbounded or heavy-tailed data, where sub-Gaussian or uniformly bounded assumptions are often violated.
> > >
> > > On the empirical side, we have strengthened the numerical evidence beyond the original ridge-regression study. In addition to the original experiments, we now include new neural-network experiments trained with ADAM, together with broader scaling experiments. These new results show the same two-regime behavior in a modern nonconvex setting and provide further evidence that the weaker moment-based assumptions are practically relevant, not merely theoretically convenient.
> > >
> > > We hope this makes the main message clear: the paper contributes a technically new proof establishing a sharp two-regime concentration theorem with the correct tail order under finite-moment stability, together with new numerical evidence showing that the theory remains meaningful beyond classical linear models. We respectfully ask the reviewer to reconsider the originality/significance assessment in light of these clarifications.
> > >
> > > Reference
> > >
> > > [1] Alain Celisse and Benjamin Guedj (2016). Stability revisited: new generalisation bounds for the Leave-one-Out. Arxiv
> > >
> > > [2] Shaojie Li and Yong Liu (2024). Concentration and Moment Inequalities for General Functions of Independent Random Variables with Heavy Tails. Journal of Machine Learning Research.
> > >
> > > [3] Shaojie Li, Bowei Zhu and Yong Liu (2024). Algorithmic Stability Unleashed: Generalization Bounds with Unbounded Losses. Proceedings of the 41st International Conference on Machine Learning.
> > >
> > > [4] Magda Peligrad, Hailin Sang, Yunda Zhong and Wei Biao Wu (2014). Exact Moderate and Large Deviations for Linear Processes. Statistica Sinica.

---

### Official Review · Reviewer_3DBh · 2026-03-12

**Soundness:** 4
**Presentation:** 3
**Significance:** 4
**Originality:** 4
**Overall Recommendation:** 5
**Confidence:** 2

**Summary:**

The paper derives a new concentration inequality beyond the bounded difference condition of McDiarmid's inequality. The rate of the concentration depends on the notion of $L_p$ diameter that is analogous to sub-Gaussian diameter.

**Compliance With Llm Reviewing Policy:**

Affirmed.

**Final Justification:**

I remain positive about the contribution of this work. The paper provides a concentration inequality for unbounded metric spaces, which is challenging and there exists quite only a few works in the current literature.

**Key Questions For Authors:**

I have only looked at the concentration inequality result of the paper so I have little concern for now. However, I will raise more questions as needed during the discussion period.

**Limitations:**

Yes, the authors discussed limitations adequately.

**Strengths And Weaknesses:**

**Soundness**: The submission is technically sound. Every result is supported by rigorous proof.

**Presentation**: The presentation is simple and quite easy to follow. However, I would suggest having a table of result to compare the result in Thm. 2.2 with previous results where the concentration is governed by sub-Gaussian/sub-Weibull diameters as well as the comparison with the Fuk-Nagaev inequality.

**Significance**: The contribution of new concentration inequalities is quite significant and much appreciated in the learning theory community.

**Originality**: The idea of $L_p$ diameter is novel even though the proof idea is quite similar to the Nagaev inequality which rely on the martingale decomposition of $\Delta_n$ (Eqn. (20)). Then, apply moment bounds on the unbounded components and concentration bounds on the truncated components. Most of the difficulty lies in bounding the probability of the concentration event of the truncated components $\mathcal{\widetilde P}_i(\Delta_n)$.

**Minor**: In the proof of Thm. 2.2, it took me a bit of time to realize why Eqn. (21) holds. Perhaps the authors can explain that the event $|\Delta_n|\ge z$ is a subset of the event that either the truncated sum exceeds $z$ or some component $\mathcal{P}_i(\Delta_n)$ exceeds $y$.

---

> ### Author Rebuttal · Authors · 2026-03-31
>
> We thank you for the detailed feedback and address your concerns sequentially below.
>
> **Presentation** The table below compares concentration results across sub-Gaussian/sub-Weibull diameters. We note that the Fuk-Nagaev inequality, though similar in form to ours, applies to sums of i.i.d. random variables. We will revise the manuscript accordingly and add the following table.
>
>
> To compare with prior concentration results, we write $H_i:=H_i(x_i,x_i')$ and focus on the relevant Orlicz norm.
>
> | Result | Assumption | Deviation bound |
> |---|---|---|
> | Kontorovich (2014) | $f$ is $1$-Lipschitz; $\lVert H_i\rVert_{\psi_2}<\infty$ for all $i$ |  $\mathbb{P} (\|f-\\mathbb{E}f\|>z)\leq 2\exp\left(-\frac{z^2}{2 \sum_{i=1}^n \lVert H_i\rVert_{\psi_2}^2}\right)$ |
> | Maurer--Pontil (2021) | $f$ is $1$-Lipschitz; $\lVert H_i\rVert_{\psi_1}<\infty$ for all $i$ | $\mathbb{P} (f-\mathbb Ef>z)\leq \exp\left(-\frac{z^2}{4e^2 \sum_{i=1}^n \lVert H_i\rVert_{\psi_1}^2)+2e(\max_i \lVert H_i\rVert_{\psi_1})z}\right) $ |
> | Li et al. (2024) | $\|f-f_i\|\le H_i$, $\lVert H_i\rVert_{\psi_\alpha}<\infty$, $0<\alpha\le 1$ | $\mathbb{P}(\|f-\mathbb Ef\|>z)\le \exp \left(-c_\alpha\frac{z^2}{\sum_{i=1}^n \lVert H_i\rVert_{\psi_\alpha}^2}\right)+\exp \left(-\frac{z^\alpha}{\max_i \lVert H_i\rVert_{\psi_\alpha}^\alpha}\right)$.|
> | Li et al. (2024) |$\|f-f_i\|\le H_i$, $\lVert H_i\rVert_{\psi_\alpha}<\infty$, $\alpha>1$ and $\alpha^{-1}+(\alpha^\*)^{-1}=1$ | $\mathbb{P}(\|f-\mathbb Ef\|>z)\le \exp \left(-c_\alpha\frac{z^2}{\sum_{i=1}^n \lVert H_i\rVert_{\psi_\alpha}^2}\right)+\exp \left(-\frac{z^\alpha}{\left(\sum_{i=1}^n \lVert H_i\rVert_{\psi_\alpha}^{\alpha^*}\right)^{\alpha/\alpha^\*}}\right)$ |
> | **Ours (Thm. 2.2)** | $\|f-f_i\|\le H_i$, $\lVert H_i\rVert_p<\infty$ for some $p\ge 2$ | $\mathbb{P}(\|f-\mathbb Ef\|>z)\le c_{1,p}\frac{\sum_{i=1}^n \mathbb E\|H_i\|^p}{z^p}+2\exp \left(-c_{2,p}\dfrac{z^2}{\sum_{i=1}^n \mathbb E\|H_i\|^2}\right)$ |
> | **Ours (Thm. 2.5)** | $\|f-f_i\|\le H_i$, $\lVert H_i\rVert_p<\infty$ for some $1< p< 2$|$\mathbb{P}(\|f-\mathbb Ef\|>z)\le \sum_{i=1}^n \mathbb P(\|H_i\|>\frac{z}{Q})+2\left(\frac{e Q^{p-1}\sum_{i=1}^n \mathbb{E}\|H_i\|^p}{z^p}\right)^Q $|
>
>
>
> **Minor** Thanks for the helpful suggestion. The idea is precisely as you describes: if every increment satisfies $|\mathcal{P}_i(\Delta_n)|<y$, then $\mathcal{P}_i(\Delta_n)=\tilde{\mathcal{P}}_i(\Delta_n)$ for all i, so
>     \begin{equation}
>         \Delta_n=\sum_{i=1}^n\mathcal{P}_i(\Delta_n)=\sum_{i=1}^n\tilde{\mathcal{P}_i}(\Delta_n).
>     \end{equation}
>     Hence, on the event $\{|\Delta_n|>z\}$, either the truncated sum itself must exceed $z$, or at least one increment satisfies $|\mathcal{P}_i(\Delta_n)|\geq y$, yielding Eqn.(21). We will expand the proof with additional detail in the camera-ready version.

---

> > ### Author Rebuttal · Reviewer_3DBh · 2026-04-03
> >
> > Dear Authors,
> >
> > Thank you for the response. In general I am quite pleased with the results. Hence, I will maintain my original positive rating.

---

> > > ### Author Response · Authors · 2026-04-05
> > >
> > > Thank you very much for your careful reading and for your encouraging comments. We are grateful that you found our response satisfactory, and we sincerely appreciate your thoughtful comments and time throughout the review process, which helped us improve our work.

---

### Decision · Program_Chairs · 2026-04-30

**Decision:**

Accept (regular)

**Comment:**

This paper introduces a theoretical framework for algorithmic stability requiring only a finite $L_p$ moment condition, advancing beyond classical bounded-differences assumptions. The authors derive a novel concentration inequality featuring two-regime tail behavior, successfully applying it to empirical risk minimization, transductive regression, and meta-learning.Despite split initial scores , the authors provided a highly effective rebuttal. They clarified the theoretical novelty of their "truncation-plus-MGF" proof technique, corrected typographical errors in the theorem statements, and added new neural network experiments trained with Adam to strengthen empirical validation. As the remaining negative scores lacked specific unresolved technical objections following the rebuttal, the committee finds the theoretical contributions to heavy-tailed learning sound and significant. I recommend a weak accept.